# Spectral Compressive Imaging via Unmixing-driven Subspace Diffusion Refinement

**Haijin Zeng**[1,*] **Benteng Sun**[2,*] **Yongyong Chen**[2,†] **Jingyong Su**[2,†] **Yong Xu**[2]
[1] Harvard University, [2] Harbin Institute of Technology (Shenzhen)
`haijin.zeng2018@gmail.com, SMARK2019@outlook.com`

## Abstract

Spectral Compressive Imaging (SCI) reconstruction is inherently ill-posed because a single observation admits multiple plausible reconstructions. Traditional deterministic methods struggle to effectively recover high-frequency details. Although diffusion models offer promising solutions to this challenge, their application is constrained by the limited training data and high computational demands associated with multispectral images (MSIs), making direct diffusion training impractical. To address these issues, we propose a novel Predict-and-unmixing-driven-Subspace-Refine framework (PSR-SCI). This framework begins with a light-weight predictor that produces an initial, rough estimate of the MSI. Subsequently, we introduce a unmixing-driven reversible spectral embedding module that decomposes the MSI into subspace images and spectral coefficients. This compact representation facilitates the adaptation of pre-trained RGB diffusion models and focuses refinement processes on high-frequency details, thereby enabling efficient diffusion generation with minimal MSI data. Additionally, we design a high-dimensional guidance mechanism enforcing SCI consistency during sampling. The refined subspace image is then reconstructed back into an MSI using the reversible embedding, yielding the final MSI with full spectral resolution. Experimental results on the standard KAIST and zero-shot datasets NTIRE, ICVL, and Harvard show that PSR-SCI enhances overall visual quality and delivers PSNR and SSIM results competitive with state-of-the-art diffusion, transformer, and deep-unfolding baselines. This framework provides a robust alternative to traditional deterministic SCI reconstruction methods. Code and models are available at `https://github.com/SMARK2022/PSR-SCI`.

## 1 Introduction

Multispectral imaging extends beyond the visible light spectrum, capturing image data across diverse wavelength ranges, such as infrared and ultraviolet spectra. This method, aided by filters or specialized instruments, reveals information beyond human perception, which is limited to red, green, and blue wavelengths. Consequently, multispectral images (MSIs) find applications in diverse fields such as remote sensing Yuan et al. (2017); Zeng et al. (2020), medical imaging Lu & Fei (2014); Meng et al. (2020b), and environmental monitoring Thenkabail et al. (2014).

Despite their utility, traditional multispectral imaging suffers from prolonged acquisition times due to spatial or temporal scanning, posing a significant hurdle for many computer vision applications Arad et al. (2022). Recent advancements in snapshot compressive imaging (SCI) systems have streamlined the acquisition of two-dimensional measurements of MSIs, facilitating efficient multispectral image acquisition and processing Cao et al. (2016); Yuan et al. (2015); Ma et al. (2021). However, SCI reconstruction poses unique challenges compared to traditional denoising or reconstruction tasks, as it must recover MSIs from compressed measurements. This process also involves coping with severe degradation caused by physical modulation, spectral compression, and unpredictable system noise.

Reconstructing MSI with full spatial-spectral resolution from a single measurement presents an inherently challenging and ill-posed inverse problem. Current methods face obstacles in accurately reconstructing specific aspects due to inadequate sampling in certain areas. Insufficient sampling

---

*Equal contribution.
†Corresponding author.

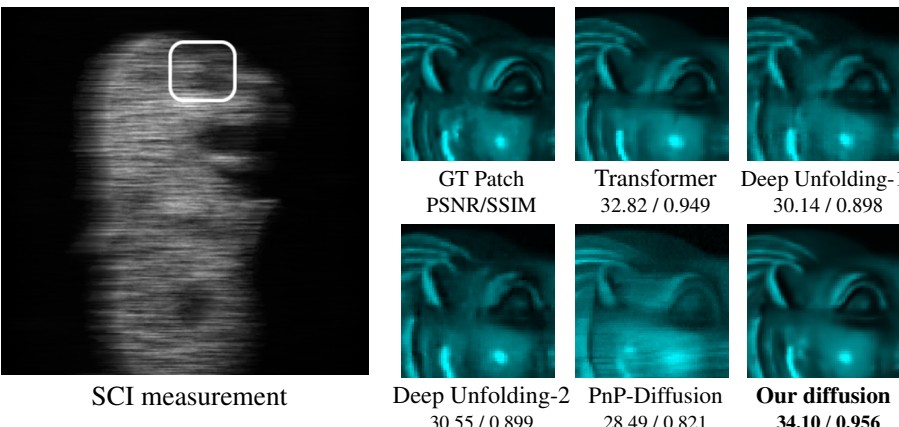

**Figure 1:** State-of-the-art methods *vs.* our PSR-SCI for snapshot compressive imaging. Transformer: CST++ Cai et al. (2022b), deep unfolding-1: GAP-Net Meng et al. (2023), deep unfolding-2 Ma et al. (2019), PnP-Diffusion Pan et al. (2024).

hinders the accurate recovery of detailed information. Specifically, contemporary end-to-end (E2E) models Meng et al. (2020a); Hu et al. (2022) are commonly trained using simulated measurement-full spatial-spectral image pairs through supervised learning. The prevailing approach involves minimizing $L_1$ or $L_2$ pixel loss, optimizing for the widely-used peak signal-to-noise ratio (PSNR) metric. However, PSNR and similar distortion metrics only partially align with human perception Blau & Michaeli (2018); Delbracio et al. (2021), sometimes resulting in visibly lower image quality in reconstructed images. To address this limitation, recent works have introduced additional loss terms Mechrez et al. (2019) aimed at enhancing image quality under metrics that more reliably represent human perception. Training networks from compressed or corrupted images to known ground truth in a supervised manner falls under the umbrella of end-to-end methods Ongie et al. (2020). While these methods perform well within their distribution, they may exhibit fragility to distributional shifts or changes in the image degradation or imaging process Jalal et al. (2021).

Diffusion model Nichol & Dhariwal (2021); Choi et al. (2021); Kawar et al. (2022) has demonstrated notable proficiency in generating content from RGB images Zhu et al. (2023). Leveraging its generative capacity to address challenging-to-reconstruct segments holds promise for enhancing multispectral SCI results Ho et al. (2020); Song et al. (2020a); Choi et al. (2021); Anderson (1982); Chung et al. (2022). Nonetheless, two significant challenges must be confronted: (i) Due to the broader spectrum captured by MSI, there is limited training data available for MSIs compared to RGB images. (ii) The high-dimensional nature of MSIs significantly increases the computational cost for diffusion denoising, especially when considering the number of sampling steps involved. Consequently, training a diffusion model directly on MSIs presents a considerable challenge.

Diffusion models pre-trained on large RGB datasets hold great potential for MSI reconstruction. However, several key challenges emerge when integrating diffusion models into the MSI domain: (1) Directly inputting MSIs, which comprise dozens of spectral bands, into existing diffusion models pre-trained on 3-channel RGB images is unfeasible due to the mismatch in channel numbers. (2) MSIs exhibit a significantly different wavelength spectrum compared to RGB images, and there exists a complex spectral interrelation among the bands of MSIs. (3) Diffusion models require considerable sampling time, a challenge intensified in MSIs by the increased computational cost of denoiser networks multiplied by sampling steps. This paper addresses these issues with four contributions:

**(i)** Our approach introduces a spectral unmixing-driven predict-and-subspace refine strategy (PSR-SCI) for SCI reconstruction. This method yields improved perceptual quality than deterministic methods and more efficient enhancement than typical diffusion models.

**(ii)** Given the ill-posedness of spectral unmixing models, we introduce a reversible decomposition module. The module performs hierarchical low-rank decomposition, preserving reversibility and exploiting spectral sparsity for compression.

**(iii)** Rather than directly enhancing the MSI, we focus the diffusion generation exclusively on the high-frequency component. This approach accelerates fine-tuning and significantly reduces the amount of required training data, thus addressing MSI data scarcity.

**(iv)** We introduce a high-dimensional guidance with SCI imaging consistency.

We evaluated the PSR-SCI performance on simulated and real datasets. As shown in Fig. 1, PSR-SCI preserves finer details and attains a higher PSNR than current SOTAs.

## 2 RELATED WORKS

The existing frameworks for SCI reconstruction predominantly consists of *model-based, Plug-and-Play, End-to-end (E2E)*, and *Deep unfolding methods. Model-based methods* Wagadarikar et al. (2008); Kittle et al. (2010); Liu et al. (2019); Wang et al. (2016); Zhang et al. (2019); Yuan (2016); Tan et al. (2016); Figueiredo et al. (2007) depend on hand-crafted image priors such as total variation, sparsity, and low-rank structures. Although these methods offer theoretical guarantees and interpretability, they require manual parameter tuning, which slows down the reconstruction process. Additionally, they are often limited by their representation capacity and generalization ability. *Plug-and-play (PnP)* algorithmsChan et al. (2016); Qiao et al. (2020); Yuan et al. (2020); Meng et al. (2021); Zheng et al. (2021b); Yuan et al. (2021b) incorporate pre-trained denoising networks into traditional model-based methods for multispectral imaging (MSI) reconstruction. However, because these pre-trained networks are fixed and not re-trained, their performance is limited by the fixed denoiser capacity and mismatch to MSI statistics.

*End-to-end (E2E) algorithms* Meng et al. (2020b;a); Hu et al. (2022); Miao et al. (2019); Yuan et al. (2021a) leverage convolutional neural networks (CNNs) to establish a mapping function from measurements to MSIs. Despite the advantages of deep learning, these methods often neglect the fundamental principles of SCI systems and are deficient in theoretical foundations, interpretability, and adaptability due to variations in imaging models. *Deep unfolding methods* Wang et al. (2020; 2019); Meng et al. (2023); Ma et al. (2019); Huang et al. (2021); Fu et al. (2021); Zhang et al. (2022), on the other hand, utilize multi-stage networks to transform measurements into MSI cubes, providing interpretability through explicit characterization of image priors and system imaging models.

In addition to the four classic frameworks mentioned above, the advancement of *generative models* Lin et al. (2023); Miao et al. (2023); Ho et al. (2020); Wang et al. (2022); Whang et al. (2022) has led to the emergence of two additional works. These works primarily aim to enhance the accuracy of SCI reconstruction by leveraging the potential of *denoising diffusion models*. Specifically, a model named DiffSCI Pan et al. (2024) utilizes a pre-trained denoising diffusion model for RGB images as the denoiser within the PnP framework. This approach combines structural insights from deep priors and optimization-based methodologies with the generative capabilities of contemporary denoising diffusion models. Another work is to use latent diffusion model to generate clean image priors for deep unfolding network, to facilitate high-quality hyperspectral reconstruction Wu et al. (2023).

## 3 OUR PSR-SCI METHOD

### 3.1 PROBLEM DEFINITION AND CHALLENGES

**Degradation Model of CASSI:** A type of snapshot compressive imaging system is the Coded Aperture Snapshot Spectral Compressive Imaging (CASSI) system Wagadarikar et al. (2008); Meng et al. (2020a); Gehm et al. (2007) shown in Fig. 2. In this system, two-dimensional measurements $\mathcal{Y} \in \mathbb{R}^{H \times (W + d \times (B-1))}$ are modulated from a three-dimensional MSI $\mathcal{X} \in \mathbb{R}^{H \times W \times B}$, where $H$, $W$, $d$, and $B$ denote the MSI's height, width, shifting step, and total number of wavelengths, respectively.

To formulate the imaging process, we firstly denote the vectorized measurement as $\mathbf{y} \in \mathbb{R}^n$ with $n = H(W + d(B-1))$ Cai et al. (2022d); Ma et al. (2019), vectorized shifted MSI as $\mathbf{x} \in \mathbb{R}^{nB}$, mask as $\mathbf{\Phi} \in \mathbb{R}^{n \times nB}$. Then, the imaging process can be formulated as:

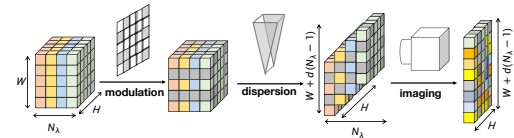

Figure 2: Illustration of a single disperser CASSI system.

$$\mathbf{y} = \mathbf{\Phi}\mathbf{x} + \mathbf{n}, \tag{1}$$

where $\mathbf{n} \in \mathbb{R}^n$ denotes the imaging noise generated by the detector. Subsequently, it is necessary to decode the measurement $\mathbf{y}$ to obtain $\mathbf{x}$ with full spatial-spectral resolution, given $\mathbf{\Phi}$ Tropp & Gilbert (2007); Donoho (2006); Jalali & Yuan (2019).

**Denoising Diffusion Models for SCI?** In addressing the inherently ill-posed nature of SCI reconstruction, existing approaches face various challenges in achieving accurate detail reconstruction simultaneously. One promising solution to this predicament lies in the denoising diffusion model, renowned for its generative capability. Nevertheless, (i) the existing diffusion-based methods are

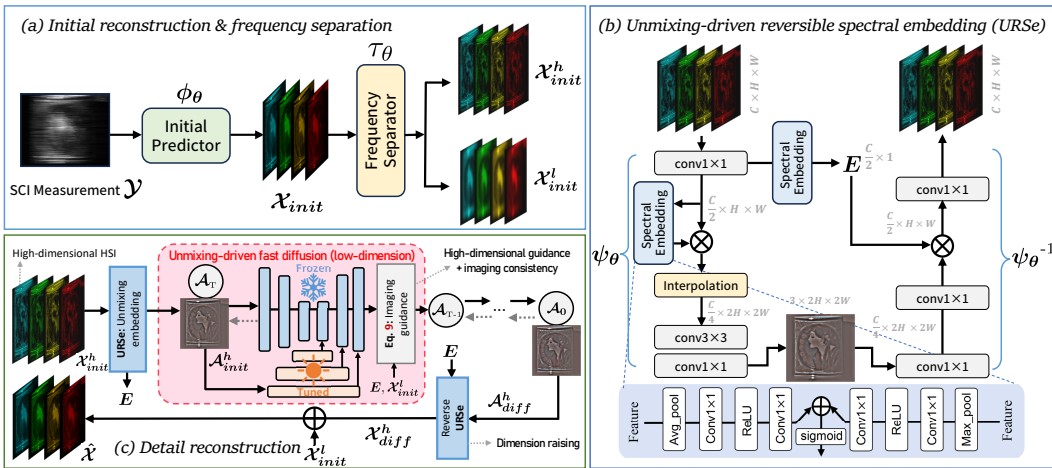

Figure 3: The overall framework of our PSR-SCI consists of three distinct yet interrelated modules, including (a) the initial predictor with frequency separator, and (b) the spectral unmixing-driven hierarchical spectral embedding, serving as a latent space decomposition method with physical significance in the context of SCI. Additionally, we (c) fine-tune the diffusion generation of high-frequency subspace images atop large-scale RGB images pre-trained models.

mostly designed for RGB images in which the input and output are with three channels, while the task of SCI reconstruction involves decoding a complete multi-band MSI from a single-band measurement. (ii) Meanwhile, limited by the inadequate datasets of MSI and the high dimension of data, the resource consumption required for retraining a powerful diffusion model from scratch on MSIs is a challenge. (iii) Furthermore, although many recent works have explored alternative sampling strategies that reduce the number of sampling steps Song et al. (2021); San-Roman et al. (2021); Kong & Ping (2021); Lee et al. (2021) for low-dimensional RGB images, the iterative diffusion process for high-dimensional MSIs with multi-bands is still time-intensive.

## 3.2 PREDICT-AND-UNMIXING-DRIVEN DIFFUSION FRAMEWORK

In this section, given a measurement $\mathcal{Y} \in \mathbb{R}^{H \times (W+d \times (B-1))}$, we introduce a method for generating a refined approximation of full spatial-spectral resolution MSI, denoted as $\hat{\mathcal{X}} \in \mathbb{R}^{H \times W \times B}$, through a *predict-and-subspace refine framework* with diffusion generation adjustment. The overall diagram of our PSR-SCI method is shown in Fig. 3. Initially, we obtain a cost-effective initial estimate via a cheap predictor $\phi_\theta$: $\mathcal{X}_{init} = \phi_\theta(\mathcal{Y})$. Then, we separate the

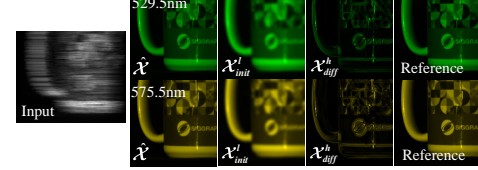

Figure 4: Illustration of initial low-frequency prediction and final high-frequency component generated from diffusion, where $\mathcal{X}^h_{diff} = \psi_\theta^{-1}(\mathcal{A}^h_{diff}, E)$.

frequency components via a frequency separator $\tau_\theta$ as depicted in Fig. 3-(a): $(\mathcal{X}^h init, \mathcal{X}^l init) = \tau_\theta(\mathcal{X}_{init})$, preserving the PSNR-critical low-frequency structures intact, while leaving the sparse, detail-rich high-frequency texture regions to the diffusion model.

Subsequently, as shown in Fig. 3-(c), to facilitate a fast diffusion process while making full use of diffusion models pre-trained by large-scale RGB data, we decompose $\mathcal{X}^h_{init}$ into low-dimensional abundance map $\mathcal{A}$ and spectral coefficient $E$ using a reversible spectral embedding module $\psi$:

$$(\mathcal{A}^h_{init}, E) = \psi_\theta(\mathcal{X}^h_{init}), \tag{2}$$

where the inverse of $\psi$, denoted as $\psi^{-1}$, satisfies that $\psi_\theta^{-1}(\mathcal{A}^h_{init}, E) \approx \mathcal{X}^h_{init}$, $\theta$ denotes the weight within the predictor and module. Subsequently, a fine-tuned diffusion model operates on this low-dimensional abundance map: $\mathcal{A}_{diff} = \mathrm{diff}(\mathcal{A}_{init})$.

To ensure the diffusion sampling process aligns with the provided measurement $\mathcal{Y}$, we modify the diffusion model to enhance the high-frequency component of $\mathcal{A}$: $\mathcal{A}^h_{diff} = \mathrm{diff}(\mathcal{A}^h_{init})$. This modification allows the fine-tuned RGB pretrained diffusion model to focus solely on modeling the residuals, thereby minimizing deviations from the measurement. Finally, we get the reconstructed MSI by reversing the spectral embedding $\psi$:

$$\hat{\mathcal{X}} = \psi_\theta^{-1}(\mathcal{A}^h_{diff}, E) + \mathcal{X}^l_{init}, \mathcal{A}^h_{diff} = \mathrm{diff}(\mathcal{A}^h_{init}). \tag{3}$$

The initial predictor, which runs only once, effectively reduces the computational burden on the subsequent diffusion model by offloading the majority of the processing tasks to itself. Our predict- and subspace refine method not only reduces the number of images required for fine-tuning the denoising diffusion process but also enables MSI generation capability through pre-trained diffusion models. Fine-tuning the RGB pre-trained denoising diffusion model with added parallel UNet encoder layers in the subspace allows for efficient diffusion sampling on high-dimensional MSI. Without this subspace sampling approach, the computational budget for iterative denoising of high-dimensional MSI increases significantly, as any rise in computational cost due to dimensionality amplifies with the number of sampling steps used.

### 3.3 Unmixing-Driven Reversible Spectral Embedding

The spectral unmixing theory posits that an MSI can be decomposed into an abundance map and spectral endmembers. It is inherently an ill-posed problem with numerous potential solutions. Abundance fractions denote the relative proportions of distinct pure materials, known as endmembers, present within a mixed pixel Keshava & Mustard (2002).

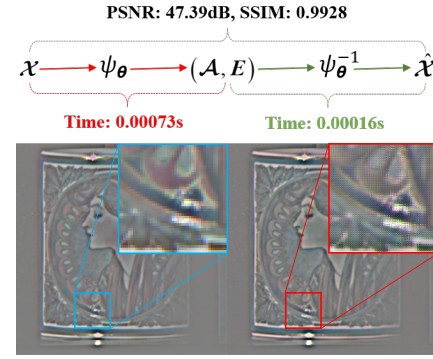

To expedite the diffusion process and leverage pre-trained RGB denoising diffusion models efficiently, we propose decomposing the underlying MSI into a reduced low-dimensional image $\mathcal{A}$ and spectral coefficients $E$ while ensuring an approximately reversible decomposition process. To achieve this, we introduce a Unmixing-driven reversible spectral embedding module (URSe). Utilizing a hierarchical spectral subspace learning strategy, as illustrated in Fig. 3-(b), URSe ensures that the compression and reconstruction gap within each stage is minimized. The backbone of URSe comprises simple $\mathrm{Conv}\, N \times N$ layers, focusing on compressing and decompressing spectral information. The upsampling operator utilized in URSe is "*Bilinear interpolation + Conv*" instead of the widely used transposed convolution to reduce the checkerboard artifacts as shown in Fig. 5.

Figure 5: Illustration of the proposed spectral embedding (top), the PSNR and SSIM are the averaged results of 10 scenes of the KAIST dataset, and comparison of upsampling within URSe (bottom).

Additionally, to mitigate information loss during the reverse process of spectral embedding, we introduce a spectral attention module to generate spectral coefficient $E$ from the embedding process. This spectral coefficient is reused during reversal to enhance reconstruction fidelity as shown in Eq. equation 3. As depicted in Fig. 5, URSe trained with the CAVE dataset achieves fast spectral embedding (0.00073s) and accurate inverse reconstruction (0.00016s), yielding a PSNR of 47.39dB and SSIM of 0.9928. Notably, due to its minimal parameter count, URSe can achieve effective training and decomposition even on a single image, as demonstrated in Fig. 10-(a)(b).

### 3.4 Unmixing-Driven MSI Diffusion Refinement

The proposed unmixing-driven reversible spectral embedding module enables the transformation of a high-dimensional MSI into a reduced low-dimensional subspace image, with a promising inverse mapping for reversal. This facilitates the utilization of diffusion models pre-trained on large-scale RGB datasets to address MSI data absence issues, while also enabling fast diffusion process to alleviate computational budget constraints for MSI.

On the basis of Sec. 3.3, this section outlines a methodology for producing accurate high-frequency subspace approximations ($\mathcal{A}_{diff}^{h}$). This is achieved by fine-tuning the stable diffusion model Rombach et al. (2022) pre-trained on large-scale RGB datasets, augmented with a tailored high-dimensional MSI control mechanism, atop the IRControlNet architecture Lin et al. (2023), as shown in Fig. 3-(c). As stable diffusion, all the diffusion processes of our method are performed in latent space, where an autoencoder Kingma & Welling (2013) is used to convert an image $x$ into a latent $z$ with encoder $\mathcal{E}$ and reconstructs it with decoder $\mathcal{D}$.

**Basic Diffusion Process**. The forward process is a Markov chain, where Gaussian noise with variance $\beta_t \in (0, 1)$ at time $t$ is progressively added to the latent $z = \mathcal{E}(x)$ to produce the noisy latent:

$$z_t = \sqrt{\bar{\alpha}_t}z + \sqrt{1 - \bar{\alpha}_t}\epsilon, \tag{4}$$

---

**Algorithm 1** Predict-and-subspace-refine diffusion sampling.

---

**Require:** $\mathcal{D}$: denoiser network, $\phi_\theta$: initial predictor, $\psi_\theta$: spectral subspace learning module, $\mathcal{Y}$: SCI measurement, gradient scale $s$, $\alpha_{1:T}$: noise schedule, $\tau$: frequency separator.
1: $\mathcal{X}_{init} \leftarrow \phi_\theta(\mathcal{Y})$      ▷ Initial prediction
2: $\mathcal{X}_{init}^h, \mathcal{X}_{init}^l \leftarrow \tau_\theta(\mathcal{X}_{init})$      ▷ Frequency separating
3: $\mathcal{A}_{init}^h, E \leftarrow \psi_\theta(\mathcal{X}_{init}^h)$      ▷ Spectral subspace embedding
4: $\boldsymbol{z}_T \sim \mathcal{N}(\boldsymbol{0}, \boldsymbol{I}_d)$      ▷ Run diffusion sampling
5: **for** $t = T, \dots, 1$ **do**
6:      $\epsilon_t \sim \mathcal{N}(\boldsymbol{0}, \boldsymbol{I}_d)$
7:      $\hat{\boldsymbol{z}}_0 \leftarrow \dfrac{\boldsymbol{z}_t}{\sqrt{\bar{\alpha}_t}} - \dfrac{\sqrt{1 - \bar{\alpha}_t}\epsilon_\theta(\boldsymbol{z}_t, t, \mathcal{E}(\mathcal{A}_{init}^h))}{\sqrt{\bar{\alpha}_t}}$      ▷ Low-dimensional subspace diffusion step
8:      $\mathcal{L}(\tilde{z}_0, \mathcal{X}_{init}) = \dfrac{1}{N} \left|\left|((\psi_\theta^{-1}(\mathcal{D}(\hat{\boldsymbol{z}}_0), E) + \mathcal{X}_{init}^l) - \mathcal{X}_{init}) + \mathcal{Y} - \Phi(\psi_\theta^{-1}(\mathcal{D}(\hat{\boldsymbol{z}}_0), E) + \mathcal{X}_{init}^l)\right|\right|_2^2$
9:      Sample $\boldsymbol{z}_{t-1}$ from $q(\boldsymbol{z}_{t-1}|\boldsymbol{z}_t, \hat{\boldsymbol{z}}_0 - s\nabla_{\boldsymbol{z}_t}\mathcal{L}(\hat{\boldsymbol{z}}_0, \mathcal{X}_{init}))$ via Eq. equation 20
10:      **return** $\mathcal{A}_{diff}^h = \mathcal{D}(\boldsymbol{z}_0)$      ▷ VAE's decoder
11: **end for**
12: **return** $\hat{\mathcal{X}} = \psi_\theta^{-1}(\mathcal{A}_{diff}^h, E) + \mathcal{X}_{init}^l$      ▷ Return MSI via reversed spectral embedding

---

where $\epsilon \sim \mathcal{N}(0, \mathbf{I})$, $\alpha_t = 1 - \beta_t$ and $\bar{\alpha}_t = \prod_{s=1}^t \alpha_s$. Subsequently, for denoising step, we train a UNet denoiser $\epsilon_\theta$ to predict the noise $\epsilon$ with randomly sampled $t$, by optimizing following loss:

$$\mathcal{L} = \mathbb{E}_{\boldsymbol{z}, \mathcal{X}_{init}^l, \mathcal{X}_{init}, t, \epsilon, \mathcal{E}(\mathcal{A}_{init}^h)}[||\epsilon - \epsilon_\theta(\sqrt{\bar{\alpha}_t}\boldsymbol{z} + \sqrt{1 - \bar{\alpha}_t}\epsilon, \mathcal{X}_{init}, \boldsymbol{\Phi}, \mathcal{Y}, \mathcal{X}_{init}^l, t, \mathcal{E}(\mathcal{A}_{init}^h))||_2^2]. \quad (5)$$

In addition, to make full use of diffusion model pre-trained large-scale RGB datasets for our MSI task, we adopt Stable Diffusion 2.1-base[1] as our pre-trained model, and fine-tune it with multispectral dataset CAVE Park et al. (2007). In addition, as illustrated in Fig. 3-(c), we incorporate a parallel encoder alongside the original encoder of UNet, as described by Lin et al. (2023). This modification enables the diffusion model to include tuneable parameters that are specifically adapted to the small-scale MSI data. Simultaneously, it retains the foundational generative capabilities conferred by pre-training on extensive RGB datasets.

**Diffusion with high-dimensional guidance and imaging consistency.** The basic diffusion generation process operates within a subspace, while the final reconstruction of SCI occurs in the high-dimensional MSI space. Consequently, even if the diffusion models produce a high-quality image within the subspace, it does not necessarily ensure a satisfactory final reconstruction in the MSI space. To address this issue, we propose the integration of a high-dimensional guidance mechanism into the conventional sampling process. This approach aims to enhance the alignment between the subspace diffusion-generated image and the ultimate high-dimensional MSI reconstruction. Specifically, we conduct the basic diffusion process in latent space but enhance it with guidance from the original high-dimensional MSI space using our reversible spectral embedding $\psi$ and its inverse $\psi^{-1}$, with the initial prediction $\mathcal{X}_{init}$ and $\mathcal{Y}$ as a reference. At time $t$, the denoiser first predicts the noise $\epsilon_t$ of the noisy latent $z_t$. Then the predicted noise $\epsilon_t$ is removed from $z_t$ to get the clean latent $\tilde{z}_0$:

$$\epsilon_t = \epsilon_\theta(\boldsymbol{z}_t, \mathcal{X}_{init}, \mathcal{A}_{init}^l, \boldsymbol{\Phi}, \mathcal{Y}, t, \mathcal{E}(\mathcal{A}_{init}^h)), \tilde{z}_0 = \frac{\boldsymbol{z}_t - \sqrt{1 - \bar{\alpha}_t}\epsilon_t}{\sqrt{\bar{\alpha}_t}}. \quad (6)$$

The reverse process is updated as follows:

$$\boldsymbol{z}_{t-1} = \frac{1}{\sqrt{\alpha_t}}\left(\boldsymbol{z}_t - \frac{1 - \alpha_t}{\sqrt{1 - \bar{\alpha}_t}}\epsilon_\theta(\boldsymbol{z}_t, \mathcal{X}_{init}, \mathcal{A}_{init}^l, \boldsymbol{\Phi}, \mathcal{Y}, t, E, \mathcal{E}(\mathcal{A}_{init}^h))\right) + \sqrt{1 - \alpha_t}z_t, \quad (7)$$

where $z_t \sim \mathcal{N}(0, 1), t \in [T]$. As Song et al. (2020b); Rui et al. (2023), we formulate the ancestral sampling process (7) as the discretization of reverse Stochastic Differential Equations (SDE).

Together with condition $\mathcal{X}_{init}$ and the spectral coefficient $E$ as conditioning variables, we reformulate the reverse SDE concerning $\boldsymbol{z}$ as

$$d\boldsymbol{z} = \left[f(\boldsymbol{z}, t) - g_t^2 \nabla_{\boldsymbol{z}_t} \log p_t(\boldsymbol{z}_t | \mathcal{X}_{init}, \boldsymbol{\Phi}, \mathcal{Y}, E)\right] dt + g(t)d\bar{\mathbf{w}}, \quad (8)$$

---

[1]https://github.com/Stability-AI/stablediffusion

Table 1: Numerical evaluations between our PSR-SCI and SOTAs across 10 simulated scenes are presented. The table includes PSNR values (upper entry) and SSIM scores (lower entry) for each method. The best and second-best outcomes are emphasized in bold and underlined, respectively.

| Algorithms | Category | Reference | S1 | S2 | S3 | S4 | S5 | S6 | S7 | S8 | S9 | S10 | Avg |
|---|---|---|---|---|---|---|---|---|---|---|---|---|---|
| DeSCI Liu et al. (2019) | Model | TPAMI 2019 | 28.38 | 26.00 | 23.11 | 28.26 | 25.41 | 24.66 | 24.96 | 24.15 | 23.56 | 24.17 | 25.27 |
| | | | 0.803 | 0.701 | 0.730 | 0.855 | 0.778 | 0.764 | 0.725 | 0.747 | 0.701 | 0.677 | 0.748 |
| $\lambda$-Net Miao et al. (2019) | CNN | ICCV 2019 | 30.10 | 28.49 | 27.73 | 37.01 | 26.19 | 28.64 | 26.47 | 26.09 | 27.50 | 27.13 | 28.53 |
| | | | 0.849 | 0.805 | 0.870 | 0.934 | 0.817 | 0.853 | 0.806 | 0.831 | 0.826 | 0.816 | 0.841 |
| TSA-Net Meng et al. (2020a) | CNN | ECCV 2020 | 32.31 | 31.03 | 32.15 | 37.95 | 29.47 | 31.06 | 30.02 | 29.22 | 31.14 | 29.18 | 31.35 |
| | | | 0.894 | 0.863 | 0.916 | 0.958 | 0.884 | 0.902 | 0.880 | 0.886 | 0.909 | 0.861 | 0.895 |
| DIP-HSI Meng et al. (2021) | PnP | ICCV 2021 | 31.32 | 25.89 | 29.91 | 38.69 | 27.45 | 29.53 | 27.46 | 27.69 | 33.46 | 26.10 | 29.75 |
| | | | 0.855 | 0.699 | 0.839 | 0.926 | 0.796 | 0.824 | 0.700 | 0.802 | 0.863 | 0.733 | 0.803 |
| BiSRNet Cai et al. (2024) | BNN | NeurIPS 2023 | 30.95 | 29.21 | 29.11 | 35.91 | 28.19 | 30.22 | 27.85 | 28.82 | 29.46 | 27.88 | 29.76 |
| | | | 0.847 | 0.791 | 0.828 | 0.903 | 0.827 | 0.863 | 0.800 | 0.843 | 0.832 | 0.800 | 0.837 |
| HDNet Hu et al. (2022) | Transformer | CVPR 2022 | 34.96 | 35.64 | 35.55 | 41.64 | 32.56 | 34.33 | 33.27 | 32.26 | 34.17 | 32.22 | 34.66 |
| | | | 0.937 | 0.943 | 0.946 | 0.976 | 0.948 | 0.928 | 0.945 | 0.944 | 0.940 | 0.946 | |
| MST-L Cai et al. (2022a) | Transformer | CVPR 2022 | 35.30 | 36.13 | 35.66 | 40.05 | 32.84 | 34.56 | 33.80 | 32.74 | 34.37 | 32.63 | 34.81 |
| | | | 0.944 | 0.948 | 0.954 | 0.976 | 0.949 | 0.955 | 0.930 | 0.950 | 0.944 | 0.943 | 0.949 |
| MST++ Cai et al. (2022c) | Transformer | CVPR 2022 | 35.57 | 36.22 | 37.00 | _42.86_ | 33.27 | 35.27 | 34.05 | 33.50 | 36.17 | 33.26 | 35.72 |
| | | | 0.945 | 0.949 | 0.959 | 0.980 | 0.954 | 0.954 | 0.936 | 0.956 | 0.956 | 0.949 | 0.955 |
| CST-L+ Cai et al. (2022b) | Transformer | ECCV 2022 | 35.64 | 36.79 | 37.71 | 41.38 | 32.95 | 35.58 | 34.54 | 34.07 | 35.62 | 32.82 | 35.71 |
| | | | 0.951 | 0.957 | 0.965 | 0.981 | 0.957 | _0.966_ | 0.947 | _0.964_ | 0.959 | 0.949 | 0.960 |
| ADMM-Net Ma et al. (2019) | Deep Unfolding | ICCV 2019 | 34.03 | 33.57 | 34.82 | 39.46 | 31.83 | 32.47 | 32.01 | 30.49 | 33.38 | 30.55 | 33.26 |
| | | | 0.919 | 0.904 | 0.933 | 0.971 | 0.924 | 0.926 | 0.898 | 0.907 | 0.917 | 0.899 | 0.920 |
| DGSMP Huang et al. (2021) | Deep Unfolding | CVPR 2021 | 33.26 | 32.09 | 33.06 | 40.54 | 28.86 | 33.08 | 30.74 | 31.55 | 31.66 | 31.44 | 32.63 |
| | | | 0.915 | 0.898 | 0.925 | 0.964 | 0.882 | 0.937 | 0.886 | 0.923 | 0.911 | 0.925 | 0.917 |
| GAP-Net Meng et al. (2023) | Deep Unfolding | IJCV 2023 | 33.63 | 33.19 | 33.96 | 39.14 | 31.44 | 32.29 | 31.79 | 30.25 | 33.06 | 30.14 | 32.89 |
| | | | 0.913 | 0.902 | 0.931 | 0.971 | 0.921 | 0.927 | 0.903 | 0.907 | 0.916 | 0.898 | 0.919 |
| DAUHST-3stg Cai et al. (2022d) | Deep Unfolding | NeurIPS 2022 | 36.59 | 37.93 | 39.32 | 44.77 | 34.82 | 36.19 | 36.02 | 34.28 | 38.54 | 33.67 | 37.21 |
| | | | 0.949 | 0.958 | 0.964 | 0.980 | 0.961 | 0.962 | 0.950 | 0.956 | 0.963 | 0.947 | 0.959 |
| DAUHST-SP2 He et al. (2024) | Subspace prior | Information Fusion 2024 | _36.73_ | 37.76 | _39.57_ | _46.21_ | 35.08 | _36.18_ | _36.66_ | **34.59** | _39.05_ | **34.23** | _37.61_ |
| | | | _0.956_ | 0.963 | _0.970_ | _0.988_ | 0.966 | _0.969_ | 0.960 | **0.966** | _0.969_ | 0.958 | _0.966_ |
| DiffSCI Pan et al. (2024) | Diffusion | CVPR 2024 | 34.96 | 34.60 | _39.83_ | 42.65 | 35.21 | 33.12 | 36.29 | 30.42 | 37.27 | 28.49 | 35.28 |
| | | | 0.907 | 0.905 | 0.949 | 0.951 | 0.946 | 0.917 | 0.944 | 0.887 | 0.931 | 0.821 | 0.916 |
| **PSR-SCI-T** | Diffusion | Ours | 36.33 | _38.57_ | 38.09 | 42.55 | _35.43_ | 35.59 | 36.29 | 34.26 | 36.57 | 33.31 | 36.68 |
| | | | 0.953 | _0.964_ | 0.966 | 0.979 | 0.964 | 0.963 | _0.954_ | _0.959_ | 0.962 | 0.948 | 0.961 |
| **PSR-SCI-D** | Diffusion | Ours | **37.18** | **38.74** | **41.04** | **46.31** | **35.81** | **36.76** | **37.38** | _34.55_ | **39.49** | _34.10_ | **38.14** |
| | | | **0.962** | **0.968** | **0.976** | **0.988** | **0.971** | **0.972** | **0.965** | 0.955 | **0.972** | _0.956_ | **0.967** |

where $f(\boldsymbol{z}, t) = -\frac{1}{2}(1 - \alpha_t)$ and $g_t = \sqrt{1 - \alpha_t}$, $\bar{\boldsymbol{w}}$ is the reverse of the standard Wiener process. The gradient $\nabla_{\boldsymbol{z}_t} \log p_t(\boldsymbol{z}_t)$ is commonly referred to the score function of $\boldsymbol{z}_t$.

Then, we discretize the reverse SDE (8) using the form of ancestral sampling process (7):

$$\boldsymbol{z}_{t-1} = \frac{1}{\sqrt{\alpha_t}} \left( \boldsymbol{z}_t + (1 - \alpha_t) \nabla_{\boldsymbol{z}_t} \log p_t(\boldsymbol{z}_t | \mathcal{X}_{init}, \boldsymbol{\Phi}, \mathcal{Y}, E) \right) \tag{9}$$

$$\approx \frac{1}{\sqrt{\alpha_t}} \left( \boldsymbol{z}_t - \frac{1 - \alpha_t}{\sqrt{1 - \bar{\alpha}_t}} \epsilon_\theta(\boldsymbol{z}_t, t) \right) + \sqrt{1 - \alpha_t}\, \boldsymbol{z}_t - s \nabla_{\boldsymbol{z}_t} \| \mathcal{X}_{init} - (\psi_\theta^{-1}(\mathcal{D}(\hat{\boldsymbol{z}}_0), E) + \mathcal{X}_{init}^l)$$
$$+ \mathcal{Y} - \boldsymbol{\Phi}(\psi_\theta^{-1}(\mathcal{D}(\hat{\boldsymbol{z}}_0), E) + \mathcal{X}_{init}^l) \|_F, \tag{10}$$

where $s$ is gradient scale, $\hat{\boldsymbol{z}}_0 = \boldsymbol{z}_{t-1}$. At time $t$, the sampling process can be divided into two distinct components. The first component involves sampling from the parameterized distribution $p(\boldsymbol{z}_{t-1}|\boldsymbol{z}_t)$ with a fixed variance of $\sqrt{1 - \alpha_t}$. The second component adjusts the sample to maintain consistency with the initial MSI prediction constraints. Please refer to the supplementary material for a detailed explanation of the process from Eq. 8 to Eq. 9. Based on (11), (9), and the basic framework described in Sec. 3.2, we summarize the pseudocode for the modified sampling procedure in Algorithm 1.

## 4 EXPERIMENTS

**Experiment Setup.** We employ a pre-trained fast **t**ransformer model Cai et al. (2022a) and a 3-stage **d**eep unfolding model Cai et al. (2022d) as initial predictors $\phi_\theta$ for our PSR-SCI-T and PSR-SCI-D, respectively. The frequency separator $\tau_\theta$ is based on Gaussian filter (see supplementary materials for details). Due to the high dimensionality of spectral data and considerations for training performance, we train the URSe and VAE models individually. We initially train the URSe model on the GT and high-frequency portion of the simulation dataset, with a spatial size of $256 \times 256$ and 28 bands from the CAVE dataset Park et al. (2007). Subsequently, we freeze the URSe and fine-tune the VAE model. For the Diffusion model, we use the well-trained Stable Diffusion 2.1-base, and fine-tune the ControlNet model to shift the diffusion model's focus from the entire image to the high-frequency texture regions using CAVE. Similar to most existing methods Meng et al. (2020a); Hu et al. (2022); Huang et al. (2021); Cai et al. (2022d), we select 10 scenes with a spatial size of $256 \times 256$ and 28

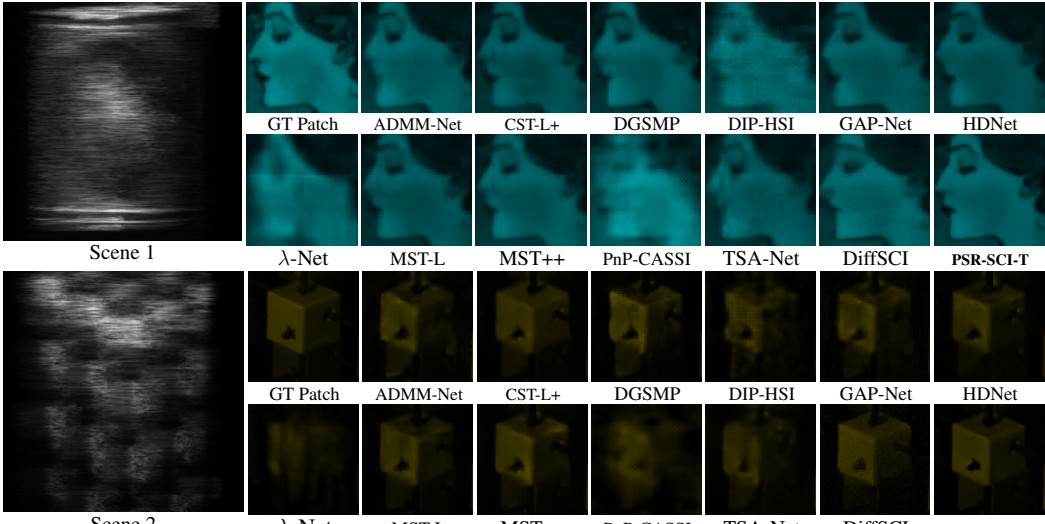

Scene 1

| GT Patch | ADMM-Net | CST-L+ | DGSMP | DIP-HSI | GAP-Net | HDNet |
| λ-Net | MST-L | MST++ | PnP-CASSI | TSA-Net | DiffSCI | **PSR-SCI-T** |

Scene 2

| GT Patch | ADMM-Net | CST-L+ | DGSMP | DIP-HSI | GAP-Net | HDNet |
| λ-Net | MST-L | MST++ | PnP-CASSI | TSA-Net | DiffSCI | **PSR-SCI-T** |

Figure 6: Visual comparison on the KAIST dataset. **Top** is $Scene$ 1 at wavelength 487.0nm. **Bottom** is $Scene$ 2 at wavelength 575.5nm. Additional KAIST results are shown in the supplemental material.

bands from KAIST Choi et al. (2017) as the simulation dataset for testing. Meanwhile, we also select 5 MSIs with a spatial size of 660×660 and 28 bands, captured by the CASSI system as the real dataset Meng et al. (2020a), and then crop the MSIs into data blocks of size 256×256 for testing.To evaluate generalization performance of our approach, we test it on several zero-shot MSI datasets, including ICVL, NTIRE, and Harvard, which were not used during training.

## 4.1 EVALUATION METRICS

We assessed our method using quantitative metrics: Peak Signal-to-Noise Ratio (PSNR) and Structural Similarity Index (SSIM). For qualitative evaluations, we analyzed local patches from both SOTA methods and our PSR-SCI method against ground truth in simulated experiments. We also compared spectral density curves of reconstructed MSIs with ground truth and calculated their correlation coefficients. In experi-

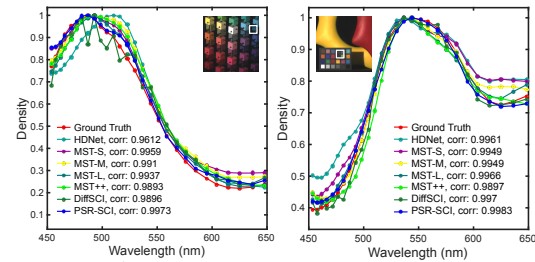

Figure 7: Spectral Density Curves.

ments with real data where MSI ground truth is absent, RGB images of the same scene served as a general reference, approximating the overall shape and details of scene objects. For zero-shot datasets, we supplemented PSNR with the MANIQA metric to thoroughly assess image fidelity and visual quality, ensuring comprehensive evaluation of our model on unseen data.

## 4.2 QUANTITATIVE RESULTS

Table. 1 and 2 show quantitative results on KAIST, ICVL, NTIRE and Harvard dataset. We compared our model with the current SOTA methods: DESCI Liu et al. (2019), λ-Net Miao et al. (2019), TSA-NET Meng et al. (2020a),

Table 2: Comparison of PSNR, SSIM, and MANIQA metrics across several zero-shot datasets.

| Dataset | Metric | DAUHST-3stg (NeurIPS 2022) | MST-L (CVPR 2022) | DPU-9stg (CVPR 2024) | SSR-L (CVPR 2024) | LADE-10stg (ECCV 2024) | DiffSCI (CVPR 2024) | PSR-SCI-D (Ours) | PSR-SCI -DPU (Ours) | PSR-SCI -SSR (Ours) |
|---|---|---|---|---|---|---|---|---|---|---|
| ICVL | PSNR↑ | 34.64 | 34.03 | 36.56 | 36.25 | 35.89 | 33.02 | 37.03 | **37.25** | 37.14 |
| | SSIM↑ | 0.890 | 0.885 | 0.918 | 0.914 | 0.904 | 0.868 | 0.918 | **0.923** | 0.918 |
| | MANIQA↑ | 0.200 | 0.209 | 0.200 | 0.209 | 0.210 | 0.207 | **0.217** | 0.216 | 0.213 |
| NTIRE | PSNR↑ | 34.44 | 33.04 | 36.25 | 35.44 | 33.58 | 32.79 | 36.44 | **36.62** | 35.53 |
| | SSIM↑ | 0.927 | 0.914 | 0.945 | 0.942 | 0.923 | 0.903 | 0.953 | **0.955** | 0.948 |
| | MANIQA↑ | 0.214 | 0.210 | 0.226 | 0.230 | 0.221 | 0.205 | 0.233 | 0.238 | **0.240** |
| Harvard | PSNR↑ | 25.57 | 24.01 | 27.05 | 25.93 | 28.02 | 24.68 | 26.90 | 28.58 | **29.02** |
| | SSIM↑ | 0.622 | 0.594 | 0.650 | 0.597 | 0.739 | 0.602 | **0.776** | 0.764 | 0.728 |
| | MANIQA↑ | 0.187 | 0.204 | 0.197 | 0.195 | 0.198 | 0.174 | 0.205 | 0.239 | **0.247** |

DGSMP Huang et al. (2021), GAP-NET Meng et al. (2023), ADMM-NeT Ma et al. (2019), PnP-CASSI Zheng et al. (2021a), DIP-MSI Meng et al. (2021), HDNET Hu et al. (2022), MST-L Cai et al. (2022a), MST++ Cai et al. (2022c), CST-L-+ Cai et al. (2022b), DiffSCI Pan et al. (2024), DPU Zhang et al. (2024a), SSR Zhang et al. (2024b) and LADE Wu et al. (2025). On KAIST, our model achieves SOTA performance across all metrics and consistently achieves the highest PSNR or SSIM scores across all 10 scenes. Specifically, we achieve an average PSNR of 36.68dB, representing an improvement of nearly 1.4dB compared to the latest SOTA method, DiffSCI, which is the current leading diffusion-based method in SCI. Furthermore, our method outperforms all transformer-based methods in terms of PSNR across all scenes except S4. These results underscore the flexibility of our framework in balancing fidelity and detail generation using a generative denoising diffusion model.

### 4.3 QUALITATIVE EXPERIMENTS

**Results on Simulation Dataset.** The detailed comparisons of local patches are presented in Fig. 6, showcasing two scenes: the $8^{th}$ band of Scene 1 (top) and the $21^{st}$ band of Scene 2 (bottom). Upon comparison with the ground truth, it is evident that our PSR-SCI method yields superior visual effects, featuring cleaner textures and fewer artifacts compared to other SOTA methods. For instance, in Scene 1, notable improvements are observed in the details of facial features such as the eyebrow, nose, and mouth. In Scene 2, a challenging scenario with dark areas, only DiffSCI and our method successfully reconstruct the complete structure of the cube. However, our PSR-SCI further refines the edges of the blocks, resulting in shapes and patterns closer to the ground truth. Furthermore, Fig. 7 displays density-wavelength spectral curves, indicating that the spectral accuracy of our model, as evidenced by the high correlation with reference curves, surpasses that of competing methods.

**Results on Real Dataset.** In addition, we evaluate the reconstruction performance of PSR-SCI on a real dataset and compare the results with the corresponding RGB image captured from the same scene, as shown in Fig. 8. From the star depicted in the figure, it is evident that PSR-SCI recovers a more complete and detailed shape with fewer artifacts compared to other SOTA methods. While other methods either produce a blurred shape or fail to reconstruct a reasonable surface for the star.

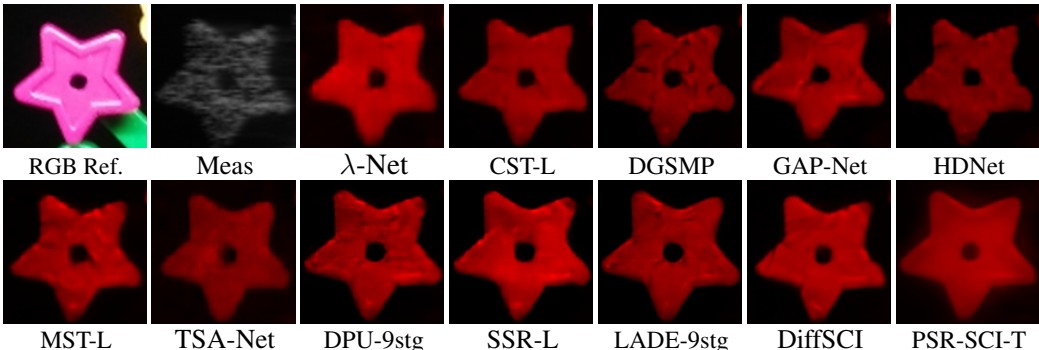

Figure 8: Visual comparison on $Scene$ 1 of real dataset at wavelength 648nm.

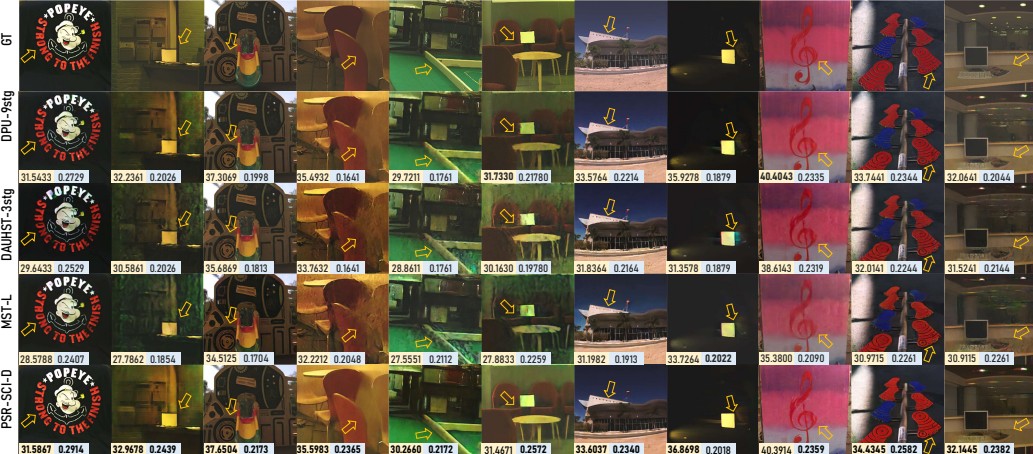

Figure 9: Comparison of methods and generalization performance testing on additional datasets (Pseudo-RGB).

**Results on Additional Datasets.** To evaluate the generalization capability of our PSR-SCI model, we conducted snapshot reconstruction tasks on several zero-shot MSI spectral datasets, including ICVL, NTIRE, and Harvard. The corresponding bands were mapped, and we compared the performance against DAUHST-3stg and MST-L. As shown in Fig. 9, the PSNR (left) and MANIQA (right) metrics are presented for each method, along with pseudo-RGB visualizations for qualitative comparison.

In addition to PSNR, we used the MANIQA metric to assess perceptual quality, providing a more comprehensive evaluation of both image fidelity and visual quality. Our PSR-SCI model consistently outperformed the competing methods across various datasets in both metrics, as summarized in Table 2. These results highlight the strong priors embedded in our diffusion-based pipeline, which enable superior zero-shot reconstruction.

These results highlight the robustness of our diffusion-based model, which leverages rich image priors to achieve superior zero-shot generation and reconstruction in MSI datasets. By consistently outperforming other methods in both traditional metrics like PSNR and perceptual quality metrics, our model demonstrates its ability to deliver enhanced reconstruction fidelity and visual accuracy across diverse zero-shot scenarios.

## 4.4 ABLATION STUDY

**Break-down Ablation.** We performed an ablation study to evaluate the impact of each component in the PSR-SCI framework. As shown in Table 3, removing the diffusion model from our framework and using only the initial predictor results in a PSNR of 37.21 dB. Using the pre-trained

Table 3: Ablation study results showing the impact of different components in PSR-SCI.

| Initial Predictor | Frequency Separator | URSe | Diffusion | PSNR↑ | SSIM↑ | LPIPS↓ | Inference Time (s) |
|---|---|---|---|---|---|---|---|
| ✓ | × | × | × | 37.21 | 0.959 | 0.05718 | 0.26 |
| × | × | × | ✓ | 33.42 | 0.883 | 0.06423 | 312.43 |
| ✓ | × | ✓ | ✓ | 36.25 | 0.940 | 0.05375 | 13.79 |
| ✓ | ✓ | × | ✓ | 37.67 | 0.962 | 0.04246 | 193.21 |
| ✓ | ✓ | ✓ | ✓ | **38.14** | **0.967** | **0.02844** | 12.90 |

diffusion model, trained on a large RGB dataset, alone results in a PSNR of 33.42 dB. While the pre-trained model possesses inherent generative capabilities, its mismatch with the spectral imaging domain leads to a performance drop. Fine-tuning the diffusion model on a spectral dataset improves performance by 2.83 dB. And the URSe module significantly reduces inference time from 312.43s to 13.79s, while frequency separator provides an additional 0.47 dB improvement. These results highlight the importance of fine-tuning, spectral embedding, and efficient high-frequency detail generation for optimal performance in spectral imaging. *In addition, the appendix A provides detailed implementation steps, optimization strategies, methodological explanations, performance analysis, dataset insights, and additional experimental results.*

**Spectral feature embedding $E$ in the URSe.** To assess the role of spectral embedding $E$, we replaced it with a constant value of 1 and retrained the model. The absence of $E$ resulted in a PSNR of 45.40 dB, compared to 49.36 dB when $E$ was included. Fig. 10 shows the decline in reconstruction quality, confirming the importance of $E$ in maintaining spectral fidelity.

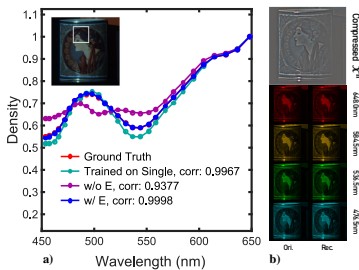

Table 4: Inference time comparison.

| Method | Category | Reference | Inference time (50 steps) | Inference time (200 steps) | Inference time (600 steps) |
|---|---|---|---|---|---|
| DiffSCI | Diffusion | CVPR 2024 | 84.54s | 251.98s | 865.81s |
| **PSR-SCI** | Diffusion | **Ours** | **8.90s** | **19.10s** | **74.76s** |

Figure 10: Spectral Reconstruction Performance of several URSe variants.

**Guidance scale $s$, time-step $T$ and inference time.** The high-dimensional guidance scale and initial time-step are critical hyperparameters in our diffusion model. We optimize these jointly, achieving the highest PSNR (36.68 dB) with $s = 0.08$ and $T = 50$ (Fig. 11). Table 4 shows our PSR-SCI model's efficiency, requiring only 8.9 seconds for 50 steps, significantly faster than the 85 seconds for the state-of-the-art DiffSCI.

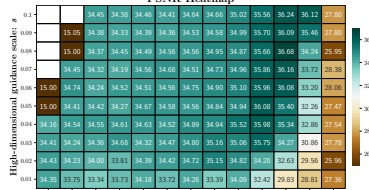

Figure 11: Hyper-parameters optimization of PSR-SCI-T.

## 5 CONCLUSION AND FUTURE DIRECTIONS

We introduced a new framework for spectral compressive imaging reconstruction, focusing on reconstructing high-frequency details by fine-tuning a diffusion model pre-trained on large-scale RGB images in the spectral subspace of MSI. To reduce the computational burden of diffusion sampling and training a diffusion model for MSI, we have proposed four novel techniques: fast SCI diffusion framework, unmixing-driven reversible spectral embedding, high-frequency diffusion generation strategy, and high-dimensional guidance with imaging consistency. Our empirical results demonstrate significant improvements in detail quality and superior metrics compared to current SOTA methods. We believe that our work introduces a novel direction in spectral compressive imaging reconstruction, emphasizing the importance of high-frequency information, and establishes a robust benchmark for future research endeavors.

Our method demonstrates excellent generalization and detail recovery capabilities. However, our approach also has certain limitations, as detailed in Appendix Sec. A.6.1. Exploring efficient denoising diffusion model sampling, faster schedulers, and enhancing our predictor and denoiser networks with optimized architectures are promising future directions.

ACKNOWLEDGMENTS

This work was supported by National Natural Science Foundation of China under Grant 62350710797, by Guangdong Basic and Applied Basic Research Foundation under Grant 2023B1515120065, by Shenzhen Science and Technology Innovation Committee under Grant JSGG20220831104402004 and Guangdong Major Project of Basic and Applied Basic Research under Grant 2023B0303000010.

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

## A  Appendix / supplemental material

**Summary.** In this supplementary material, we provide the implement details of our approach in Sec. A.1, show the optimization results of hyper-parameters: high-dimensional guidance scale $s$ and the start timestep $T$ of diffusion model by using SSIM in Sec. A.2, and provide the derivation process of the subspace diffusion with high-dimensional guidance in Sec. A.3. Section A.4 elaborates on the derivation and explanation of the Unmixing-driven Spectral Embedding (URSe) approach, which plays a crucial role in enhancing the spectral reconstruction process. Section A.6.2 discuss how the framework, rather than solely relying on parameter scaling, contributes to its superior performance. Section A.7 contrasts the differences between the generated dataset and real collected datasets, offering further insights into the data used for training. Section A.8 presents additional experimental results on both real and simulated datasets, further validating the effectiveness of our approach across different scenarios. The Table list of this supplementary materials is listed as follows:

- Sec. A.1: Implement Details
- Sec. A.2: Hyper-parameters Optimization
- Sec. A.3: Subspace Diffusion with High-dimensional Guidance
- Sec. A.4: Explanation of the Unmixing-driven Spectral Embedding Approach
- Sec. A.5: Framework Architectural Analysis and Performance Validation
- Sec. A.6: Limitation analysis and Generalization Across Diverse Datasets
- Sec. A.7: Analysis of Training Datasets for Diffusion Models
- Sec. A.8: Additional Experimental Results

## A.1 IMPLEMENT DETAILS

**More Experimental Details.** We enhance the multispectral image dataset (all from CAVE dataset https://cave.cs.columbia.edu/repository/Multispectral) using various augmentation techniques, including cropping (directly cropping to the target size and 1/2 by 1/2 cropping for $2 \times 2$ patching) and scaling ($2 \times 2$ up-scaling and $2 \times 2$ down-scaling). Additionally, during the task of real images restoration, Gaussian noise of varying degrees is introduced to the dataset to fine-tune the ControlNet module which can simulate the sensing noise in the real data. Subsequent training is conducted on the high-frequency components (with 3 Gaussian iterations) of the spectral images output by the Initial Predictor (a pre-trained fast transformer model Cai et al. (2022a)).

All experiments are conducted with data paralleling on a server equipped with 4 RTX 3090 GPUs, using Python 3.9.19, PyTorch 2.2.0+cu121, and CUDA 12.2.

1. For the URSe training, we initialize the model randomly. Training is performed using the Adam optimizer with a learning rate ($l_r$) of $0.02$ and a batchsize of 8 for 200 epochs. During the first $30\%$ of the epochs, an additional visual enhancement regularization term based on the mean squared error (MSE) between encoded images and pseudo-RGB images is included. The first 10 epochs employ a linear learning rate warmup strategy, and over the next 100 epochs, the $l_r$ is gradually reduced to $0.002$ to achieve higher performance. The initial training of URSe takes approximately 2.5 hours.

2. For the VAE module training, we fine-tune the VAE module from the well-trained SD2.1 model (https://github.com/Stability-AI/StableDiffusion). Training is conducted with $l_r = 1 \times 10^{-5}, batch\_size = 16$, and using the Adam optimizer with data parallelism across four GPUs. Similarly, a 10% step linear warmup is applied, with a total of 40,000 training steps, taking about 9.5 hours.

3. Next, we fine-tune the URSe output part by serially integrating both modules to avoid the cumulative errors that might arise from independent training. This fine-tuning is performed with $l_r = 1 \times 10^{-6}, batch\_size = 6$ for 100 steps, taking approximately 3 hours.

4. Then, we perform a full-parameter fine-tuning of the Diffusion's ControlNet part within the RGB environment converted by URSe. Training uses the Adam optimizer with a batch size of 24 and is conducted in two phases: the first phase sets $l_r = 2 \times 10^{-5}$ training for 30,000 steps, and the second phase sets $l_r = 4 \times 10^{-6}$ training for 50,000 steps, resulting in a total training time of 18 hours.

Finally, during the simulated data testing experiments , we fix the random seed to $480$(not fixed on real data to fully demonstrate the robust prior knowledge of the Diffusion model). This ensures that the output results of the Diffusion model are fixed and reproducible. The Diffusion model is used to complete the texture restoration task and subsequent testing. The hyper-parameter grid search is also conducted under the condition of the random seed set to $480$.

**Method for Image High and Low-Frequency Separation.** The core method for separating the high and low-frequency components of an image is through Gaussian low-pass filtering. We apply the same Gaussian low-pass filter (using a Gaussian convolution kernel to perform Gaussian blur on the image) iteratively to the original multispectral image. This iterative process ultimately yields the low-frequency component of the image, while the difference from the original image represents the high-frequency texture component. By controlling the number of Gaussian low-pass filtering iterations, we can effectively manage the scale of texture details restored by the diffusion model. For the $256 \times 256$ spectral task, given that the Initial Predict effectively restores the structural parts of the image, we set the number of iterations to **3** in the global task environment.

The low-frequency content retains the low-rank spectral information of the overall image, while the high-frequency component preserves the image's texture details (especially sharp edges). By decomposing the image into high and low-frequency components, we can easily control the scale of the texture details during the diffusion restoration steps. This approach also significantly reduces the damage to the low-rank spectral information caused by channel compression and the VAE part of the latent diffusion process Rombach et al. (2022).

**Up-sampling within URSe.** To upsample multispectral images from a resolution of $256 \times 256$ to $512 \times 512$, and thereby fully exploit the prior knowledge embedded in the $512 \times 512$-resolution-

trained UNet model in the diffusion model, we experiment with various upsampling techniques. These methods include, but are not limited to, naive interpolation methods and transposed convolutions, each presenting certain challenges, as shown in Table 5.

Specifically, although interpolation methods vary, naive interpolation techniques fail to enable the model to enhance spatial utilization through learnable parameters. This limitation impedes the compression and storage of more spectral information from different channels at the same spatial location, and additionally leading to a certain degree of blurring in the upsampled images inevitably, which is detrimental to the inverse operation during the recovery phase. Conversely, direct 2x2 transposed convolutions can produce the "checkerboard artifacts" similar to the Bayer Pattern (can be seen in Fig. 5). This artifact in the low-frequency components contradicts the original goal of achieving a "low-frequency controllable diffusion model" and hinders the VAE network's image representation through high-low frequency separation.

Table 5: Comparison of different up-sampling operations used in the proposed URSe.

| Up-sampling method | w/o upsampling | Nearest | Bilinear | Trans-conv | Nearest+conv | **Bilinear+conv** |
|---|---|---|---|---|---|---|
| PSNR | 39.58 | 42.12 | 44.25 | 47.16 | 47.24 | **47.39** |
| SSIM | 0.9647 | 0.9737 | 0.9821 | **0.9958** | 0.9857 | 0.9928 |

After extensive experimentation, we adopt a combination of bilinear interpolation followed by a 3x3 convolution with stride and padding both of 1. This approach not only avoids the artifacts associated with convolutional upsampling but also allows the model to fully leverage the upsampled resolution through learnable parameters. Consequently, it delivers an excellent and high-performance image upsampling operation.

## A.2 HYPER-PARAMETERS OPTIMIZATION USING SSIM AND VISUAL MAP

As highlighted in our paper, the high-dimensional guidance scale $s$ and starting time-step $T$ of the diffusion model are critical hyper-parameters. Initially, we optimize them jointly based on PSNR. Additionally, we perform optimization based on SSIM, as depicted in Fig. 12. The trends of the parameters with respect to SSIM closely resemble those with respect to PSNR, where the best values for $s$ and $T$ are: $s = 0.08$, $T = 50$, respectively.

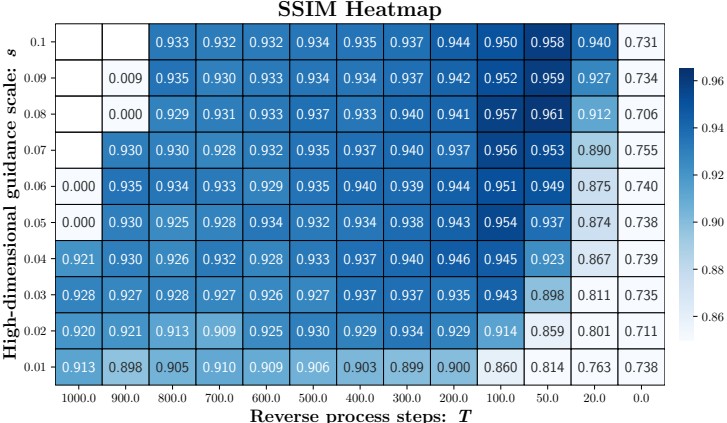

Figure 12: Hyper-parameters of PSR-SCI-T: guidance scale $s$ and time-step $T$ optimization by using SSIM.

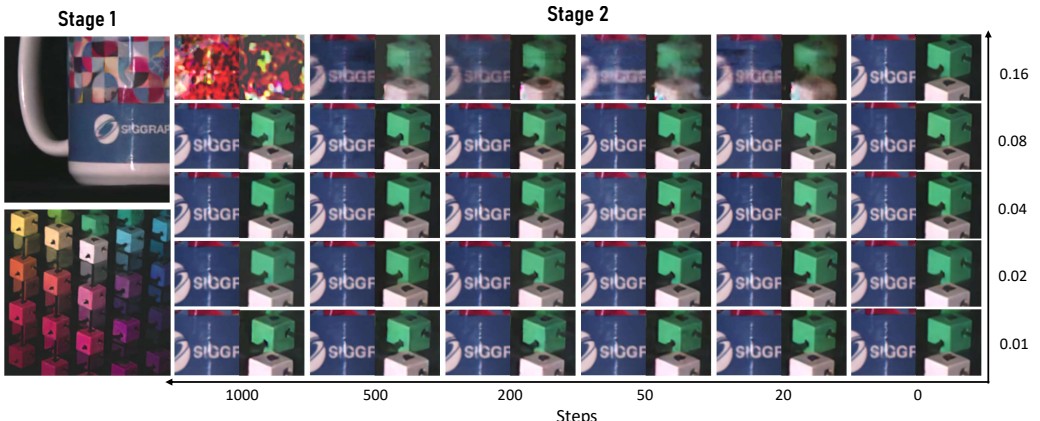

Figure 13: Visual comparison of hyper-parameters

Notably, regardless of whether PSNR or SSIM is used as the optimization metric, we found that an excessively high guidance scale ($s$) results in oscillations and amplification of the Guidance Loss during the guidance iterations, ultimately causing gradient overflow and image content collapse. This instability is highlighted in Fig. 13, where it is evident that Stage 2 significantly enhances high-frequency details compared to Stage 1. However, overly high guidance scaling can interfere with the diffusion process, leading to image degradation, while a very low guidance scale causes mismatches between high and low-frequency details. We hypothesize that this phenomenon is due to the excessive guidance scaling factor, which competes with the diffusion process, leading to image content degradation.

Through hyper-parameter search, we prove the necessity of the guidance step(Eq.( 20)) in restoring image textures during diffusion. Without adequate guidance (either a very small $g_s cale$ or insufficient $g_s teps$), the randomness inherent in the diffusion process leads to uncontrollable texture content, resulting in significantly lower restoration performance.

### A.3 ADDITIONAL DETAILS OF SUBSPACE DIFFUSION WITH HIGH-DIMENSIONAL GUIDANCE

Due to space limitations, the main body of our manuscript focuses on the essential procedures of the guided reverse denoising diffusion process. This section provides additional comprehensive details.

The diffusion generation process is performed within a lower-dimensional subspace, while the final reconstruction occurs in the high-dimensional multi-spectral image space. However, achieving high-quality images in the subspace does not always guarantee satisfactory reconstructions in the MSI space. To address this limitation, we have integrated a high-dimensional guidance mechanism into the sampling process. This enhancement ensures better alignment between subspace images and high-dimensional MSI reconstructions.

Specifically, our approach involves conducting the diffusion process in the latent space. Firstly, our approach leverages a diffusion process that transitions through a series of states, ultimately refining the latent representation. This is achieved through an iterative reverse process as detailed in equation 11 and equation 12, enabling precise sampling from the target distribution. Then, by interpreting the diffusion and reverse processes as solutions to Stochastic Differential Equations (SDEs), we align our method with a rigorous mathematical framework, as detailed in equation 13 to equation 15. This perspective highlights the relationship between continuous-time diffusion and the discretized reverse sampling steps, ensuring that our approach effectively captures the underlying data distribution while facilitating high-quality image reconstruction. Consequently, in the context of SCI reconstruction, we leverage the learned distribution of latent variables from a diffusion model to reconstruct the high-frequency components of subspace images, incorporating prior knowledge for improved accuracy. By using the initial prediction, measurement matrix, observed data, and embedding operators as guidance, we establish a connection between the subspace and MSI space, ensuring coherence and alignment during the reconstruction process. This reformulated approach enhances the reverse process and effectively bridges the gap between latent space and high-dimensional MSI reconstruction. This customized diffusion process is guided by the high-dimensional MSI space through our reversible spectral embedding functions, $\psi$ and its inverse, $\psi^{-1}$. These functions establish a connection between

the latent space and the MSI space, ensuring information is effectively transferred during sampling. Additionally, we incorporate the initial prediction $\mathcal{X}_{init}$ and the real measurement $\mathcal{Y}$ as references during the sampling phase, as shown in equation 16 to equation 20. These inputs provide essential guidance, aligning the latent space diffusion process with the high-dimensional reconstruction requirements.

By integrating this high-dimensional guidance mechanism, we improve the coherence between the generated subspace images and the final MSI reconstructions. This ensures that the diffusion process is informed by high-dimensional data, leading to more accurate and reliable results.

Specifically, at time $t$, the denoiser first predicts the noise $\epsilon_t$ of the noisy latent $z_t$. Then the predicted noise $\epsilon_t$ is removed from $z_t$ to get the clean latent $\tilde{z}_0$:

$$\epsilon_t = \epsilon_\theta(\boldsymbol{z}_t, \mathcal{X}_{init}, \mathcal{A}_{init}^l, t, \mathcal{E}(\mathcal{A}_{init}^h)), \tilde{z}_0 = \frac{\boldsymbol{z}_t - \sqrt{1 - \bar{\alpha}_t}\epsilon_t}{\sqrt{\bar{\alpha}_t}}. \tag{11}$$

Consequently, a more precise image can be sampled at state $z_0$ through an iterative reverse process denoted as $p(\boldsymbol{z}_0|\boldsymbol{z}_t)$. As mentioned in the main body of our paper, the reverse process is updated as follows:

$$\boldsymbol{z}_{t-1} = \frac{1}{\sqrt{\alpha_t}}\left(\boldsymbol{z}_t - \frac{1 - \alpha_t}{\sqrt{1 - \bar{\alpha}_t}}\epsilon_\theta(\boldsymbol{z}_t, \mathcal{X}_{init}, \mathcal{A}_{init}^l, t, \mathcal{E}(\mathcal{A}_{init}^h))\right) + \sqrt{1 - \alpha_t}z_t, \tag{12}$$

where $z_t \sim \mathcal{N}(0, 1), t \in [T]$. As Song et al. (2020b); Rui et al. (2023), we formulate the ancestral sampling process (12) as the discretization of reverse SDE.

$$dA = \left[f(A, t) - g_t^2 \nabla_{\boldsymbol{z}_t} \log p_t(\boldsymbol{z}_t)\right] dt + g_t d\bar{\mathbf{w}}, \tag{13}$$

Recent work Song et al. (2020b) shows that as the total diffusion step "$T$" goes infinity and the forward series $\{X_t\}_{t=1}^T$ becomes $\{X_t | t \in [0,1]\}$ indexed by continuous time variable, the diffusion process $X_t$ is actually the solution to an Itô SDE: $dX = f(X, t)dt + g_t d\mathbf{w}$, where $\mathbf{w}$ represents the standard Wiener process. For example, the diffusion process with transition distribution $q(X_t | X(t-1)) = \mathcal{N}(X_t | \sqrt{\alpha_t} X_{t-1}, (1 - \alpha_t)I)$ corresponds to the SDE as follows

$$dX = -\frac{1}{2}(1 - \alpha_t)dt + \sqrt{1 - \alpha_t}d\mathbf{w}. \tag{14}$$

In this case, $f(X, t) = -\frac{1}{2}(1 - \alpha_t)$ and $g_t = \sqrt{1 - \alpha_t}$. Also, the reverse process is a solution to an SDE:

$$dX = \left[f(X, t) - g_t^2 \nabla_{X_t} \log p_t(X_t)\right] dt + g_t d\bar{\mathbf{w}}, \tag{15}$$

Viewed through the lens of SDE, sampling from $p(\boldsymbol{z}_0)$ can be achieved by appropriately discretizing Equation (15). Consequently, in the context of SCI reconstruction, we aim to utilize the learned distribution of $\boldsymbol{z}$ from a diffusion model, where $\mathcal{A}_{diff}^h = \mathcal{D}(\boldsymbol{z}_0)$. This model inherently incorporates prior information of the subspace image, facilitating the reconstruction of the high-frequency component of the subspace image from the observed measurement. Then, using the initial prediction $\mathcal{X}_{init}, \mathbf{\Phi}, \mathcal{Y}$, and the $E$ (together with $\psi^{-1}$, it is used to reverse the spectral embedding and connect the subspace with the MSI space where $\mathcal{X}_{init}$ belongs) as guidance or condition, we reformulate the reverse SDE concerning $\boldsymbol{z}$ as

$$d\boldsymbol{z} = \left[f(\boldsymbol{z}, t) - g_t^2 \nabla_{\boldsymbol{z}_t} \log p_t(\boldsymbol{z}_t|\mathcal{X}_{init}, \mathbf{\Phi}, \mathcal{Y}, E)\right] dt + g_t d\bar{\mathbf{w}}, \tag{16}$$

where $f(\boldsymbol{z}, t) = -\frac{1}{2}(1 - \alpha_t)$ and $g_t = \sqrt{1 - \alpha_t}$, $\bar{\mathbf{w}}$ is the reverse of the standard Wiener process.

The gradient $\nabla_{\boldsymbol{z}_t} \log p_t(\boldsymbol{z}_t)$ is commonly referred to as the score function of $\boldsymbol{z}_t$. Using Bayes's rule, the score function can be separated into two parts

$$\nabla_{\boldsymbol{z}_t} \log p_t(\boldsymbol{z}_t|\mathcal{X}_{init}, \mathbf{\Phi}, \mathcal{Y}, E)\nabla_{\boldsymbol{z}_t} \log p_t(\boldsymbol{z}_t) + \nabla_{\boldsymbol{z}_t} \log p_t(\mathcal{X}_{init}, \mathbf{\Phi}, \mathcal{Y}, E|\boldsymbol{z}_t).$$

The first part can be derived under the general unconditional framework. However, the second part is intractable, since only the relation between $\mathcal{X}_{init}, \mathcal{X}$ and $p(\boldsymbol{z}_t|\boldsymbol{z}_0)$ are known. Following Chung et al. (2023); Rui et al. (2023), we approximate the second term as

$$\begin{aligned}\nabla_{\boldsymbol{z}_t} \log p_t(\mathcal{X}_{init}, \mathbf{\Phi}, \mathcal{Y}, E|\boldsymbol{z}_t) &= \nabla_{\boldsymbol{z}_t} \log \int p(\mathcal{X}_{init}, \mathbf{\Phi}, \mathcal{Y}, E|\boldsymbol{z}_0)p(\boldsymbol{z}_0|\boldsymbol{z}_t)d\boldsymbol{z}_0 \\ &\approx \nabla_{\boldsymbol{z}_t} \log p(\mathcal{X}_{init}, \mathbf{\Phi}, \mathcal{Y}, E|\hat{\boldsymbol{z}}_0),\end{aligned} \tag{17}$$

where $\hat{z}_0$ is the expectation of $z_0|z_t$ by Tweedie's formula:

$$\hat{z}_0(z_t) = \mathbb{E}[z_0|z_t]$$
$$= \frac{1}{\sqrt{\bar{\alpha}_t}}\left[z_t + (1-\bar{\alpha}_t)\nabla_{z_t}\log p_t(z_t)\right]. \tag{18}$$

The term $\log p(\mathcal{X}_{init}, E|\hat{z}_0)$ is much more available since $\hat{z}_0$ can be seen as an approximation to $z$, by using computational relaxation Rui et al. (2023), it can be formulated as

$$\log p(\mathcal{X}_{init}, E|\hat{z}_0) = \log p(\mathcal{X}_{init}, \Phi, \mathcal{Y}, E|\hat{z}_0)$$
$$\approx -s\|\mathcal{X}_{init} - (\psi_\theta^{-1}(\mathcal{D}(\hat{z}_0), E) + \mathcal{X}_{init}^l) + \mathcal{Y} - \Phi(\psi_\theta^{-1}(\mathcal{D}(\hat{z}_0), E) + \mathcal{X}_{init}^l)\|_F, \tag{19}$$

where $s$ is trade-off parameter. Then, we discretize the reverse SDE (16) using the form of ancestral sampling process (12):

$$z_{t-1} = \frac{1}{\sqrt{\alpha_t}}\left(z_t + (1-\alpha_t)\nabla_{z_t}\log p_t(z_t|\mathcal{X}_{init}, \Phi, \mathcal{Y}, E)\right) \tag{20}$$

$$\approx \frac{1}{\sqrt{\alpha_t}}\left(z_t - \frac{1-\alpha_t}{\sqrt{1-\bar{\alpha}_t}}\epsilon_\theta(z_t, t)\right) + \sqrt{1-\alpha_t}z_t$$
$$- s\nabla_{z_t}\|\mathcal{X}_{init} - (\psi_\theta^{-1}(\mathcal{D}(\hat{z}_0), E) + \mathcal{X}_{init}^l) + \mathcal{Y} - \Phi(\psi_\theta^{-1}(\mathcal{D}(\hat{z}_0), E) + \mathcal{X}_{init}^l)\|_F, \tag{21}$$

where $s$ is gradient scale, $\hat{z}_0 = z_{t-1}$. At this point, we have completed the detailed inference process to obtain the update step for the modified latent diffusion model using our high-dimensional guidance. $\square$

### A.4 EXPLANATION OF THE UNMIXING-DRIVEN SPECTRAL EMBEDDING APPROACH

**Motivation and Comparison with SVD-based Methods.** The motivation for using the unmixing-driven spectral embedding (URSe) approach is to effectively reduce the dimensionality of spectral information channels while ensuring accurate spectral data representation and compatibility with RGB space. Unlike SVD-based decomposition methods, such as the one used in HIR-Diff Pang et al. (2024b), which rely on static linear assumptions, our learnable URSe module captures the inherent nonlinearity of spectral data, enabling it to provide more accurate and adaptive spectral embeddings. SVD-based approaches are inherently limited in two critical aspects:

- *Forward-Decomposition Error Amplification:* SVD-based methods cannot adaptively fine-tune the reverse process, leading to amplified errors introduced during forward decomposition when reconstructing spectral images.

- *Misalignment with Spectral Characteristics:* These methods focus on rough spectral-to-RGB mapping, failing to capture the complex nonlinear relationships between spectral bands. This results in deviations between the diffusion model's outputs and true spectral images, especially given the differences in wavelength ranges (RGB: 460–650 nm vs. spectral imaging: 750–2500 nm).

Table 6 summarizes the comparison of our method, URSe, with the SVD-based decomposition on the KAIST dataset. The table includes PSNR values (upper entry) and SSIM scores (lower entry) for each method across 10 scenes. Our learnable URSe module addresses these limitations by fine-tuning on spectral datasets, ensuring accurate alignment of spectral domains with pre-trained RGB diffusion models. This capability enables task-specific spectral embeddings that are reversible, noise-robust, and optimized for high-quality SCI reconstruction. Additionally, URSe enhances both convergence and accuracy, achieving significant improvements in reconstruction speed and quality compared to traditional methods.

**Robustness of URSe in the Presence of Noise.** To further evaluate the robustness of our URSe spectral embedding module under noisy conditions, we conducted additional experiments with Gaussian noise perturbations at different levels. As shown in Table 7, the URSe module consistently maintained a high PSNR (above 37 dB) even with increasing noise levels, demonstrating its strong resilience to noise. Specifically, even with significant noise (0.1 Gaussian perturbation), URSe's PSNR dropped only slightly from 38.14 dB to 37.04 dB, while maintaining a high SSIM value.

Table 6: Comparison of reconstruction performance between SVD-based band selection (HIR-Diff) and our learnable URSe module on the KAIST dataset.

| Method | S1 | S2 | S3 | S4 | S5 | S6 | S7 | S8 | S9 | S10 | Avg |
|---|---|---|---|---|---|---|---|---|---|---|---|
| SVD-band-select (HIR-Diff) | 40.65 | 36.25 | 39.20 | 51.10 | 38.70 | 40.60 | 43.12 | 40.80 | 33.00 | 46.60 | 41.00 |
| | 0.9810 | 0.9667 | 0.9795 | 0.9948 | 0.9909 | 0.9873 | 0.9862 | 0.9867 | 0.9623 | 0.9913 | 0.9827 |
| URSe (Ours) | 50.54 | 54.03 | 49.61 | 57.02 | 49.28 | 49.99 | 47.75 | 49.05 | 50.57 | 50.97 | **50.88** |
| | 0.9982 | 0.9990 | 0.9939 | 0.9994 | 0.9984 | 0.9984 | 0.9948 | 0.9977 | 0.9967 | 0.9988 | **0.9975** |

Table 7: Ablation study on the robustness of spectral embedding for the diffusion model on the KAIST dataset under Gaussian noise perturbations.

| Method | Noise (Gaussian) Perturbation | PSNR (dB) ↑ | SSIM ↑ |
|---|---|---|---|
| URSe (Ours) | 0 | **38.14** | **0.9670** |
| | 0.01 | 38.10 | 0.9665 |
| | 0.1 | 37.04 | 0.9567 |
| SVD-band-select (HIR-Diff) | 0 | 36.87 | 0.9628 |
| | 0.01 | 36.15 | 0.9623 |
| | 0.1 | 26.82 | 0.9383 |

In contrast, the SVD-based method (HIR-Diff) showed significantly reduced performance, with a sharp drop in PSNR (from 36.87 dB to 26.82 dB) as the noise level increased. This indicates that SVD-based methods are more sensitive to noise, unable to maintain reconstruction quality under perturbations.

These results highlight the superior noise robustness of the URSe module, which is critical for real-world SCI tasks where noise is often present. The ability of URSe to retain high-quality reconstructions in noisy environments further underscores its advantages over traditional methods.

## A.5 FRAMEWORK ARCHITECTURAL ANALYSIS AND PERFORMANCE VALIDATION

**Importance of Initial Predictor in the Two-Stage Architecture**

We removed the initial predictor from the two-stage architecture and directly input the shifted-back measurement results into the URSe encoder for diffusion. Without the initial predictor, the model achieved only 24.15 dB PSNR after joint fine-tuning, significantly impairing MSI reconstruction. Incorporating the initial predictor greatly enhanced reconstruction performance (Fig. 14), highlighting its essential role in improving MSI reconstruction quality.

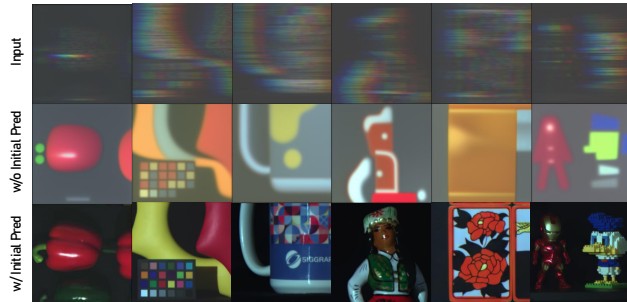

Figure 14: Visual comparison of initial predictor ablation studies.

These results demonstrate that although the diffusion model pre-trained on RGB datasets has generative capabilities, an initial prediction network or an iterative formulation like DiffSCI is still necessary to efficiently model the mapping between snapshot single-exposure images and multispectral data. This is something that diffusion models pre-trained on RGB data are not well-suited for.

**Comparison with Refinement-Based Frameworks.** To further validate the effectiveness of our approach, we conducted a detailed comparison with DAUHST-SP2 He et al. (2024), a state-of-the-art refinement-based framework. For a fair evaluation, both methods use DAUHST as the first-stage predictor. The results, summarized in Table 8, highlight the advantages of our PSR-SCI method across multiple evaluation metrics.

Specifically, our observations indicate that PSR-SCI achieves a PSNR of 38.14 dB, outperforming DAUHST-SP2's 37.61 dB. In addition, our method demonstrates superior SSIM, LPIPS, MUSIQ, MANIQA, and CLIP-IQA metrics, emphasizing its capacity to improve both objective and perceptual reconstruction quality. These improvements highlight the effectiveness of our method in enhancing reconstruction quality compared to existing refinement frameworks.

Table 8: Comparison between PSR-SCI and refinement-based frameworks. **DAUHST is used as the first-stage predictor for a fair comparison**.

| Method | Category | Reference | PSNR (dB) ↑ | SSIM ↑ | LPIPS ↓ | MUSIQ ↑ | MANIQA ↑ | CLIP-IQA ↑ |
|---|---|---|---|---|---|---|---|---|
| DAUHST-SP2 | Subspace prior | Information Fusion 2024 | 37.61 | 0.966 | - | - | - | - |
| DiffSCI | Diffusion | CVPR 2024 | 35.28 | 0.916 | 0.07421 | 39.64 | 0.23598 | 0.3315 |
| **PSR-SCI-D** | Diffusion | Ours | **38.14** | **0.967** | **0.02844** | **42.73** | **0.2527** | **0.3561** |

By including this comparison, we aim to substantiate the advantages of our approach over similar models. The consistent improvements across multiple evaluation metrics reinforce the robustness and efficacy of PSR-SCI in spectral image reconstruction.

**Comparison with HIR-Diff.** For a comprehensive comparison, we applied the HIR-Diff Pang et al. (2024b) model to the second stage of our two-stage framework while keeping the first stage unchanged. The performance results for snapshot compressive imaging tasks are presented in Table 9. Our PSR-SCI outperforms HIR-Diff in both PSNR and SSIM, demonstrating the efficacy of our proposed approach in SCI applications.

Table 9: Comparison between HIR-Diff and PSR-SCI on Snapshot Compressive Imaging (SCI) tasks.

| Method | Reference | PSNR (dB) ↑ | SSIM ↑ |
|---|---|---|---|
| HIR-Diff (+our first stage) | CVPR-2024 | 35.23 | 0.959 |
| PSR-SCI-D (Ours) | - | **38.14** | **0.967** |

These findings underline the benefits of our approach, where fine-tuning, spectral embedding, and efficient high-frequency detail generation significantly improve both the quality and speed of spectral image reconstruction from compressed sensing measurements.

Comparison with HIR-Diff for Snapshot Compressive Imaging (SCI). Our work differs from HIR-Diff in several important aspects, as summarized below:

- Spectral Decomposition: We employ a learnable spectral embedding approach, whereas HIR-Diff uses a fixed 3-band selection. Our method achieves superior reconstruction quality (50.88 dB vs. 41 dB, averaged over 10 KAIST scenes) and faster processing speed (0.0009s vs. 0.001215s). These improvements are indicative of the advantages of our trainable spectral decomposition.

- Diffusion Refinement: Unlike HIR-Diff, which applies a pre-trained RGB diffusion model without addressing the spectral differences between RGB and multispectral images, our framework incorporates a trainable diffusion model fine-tuned on spectral data. This allows our model to achieve higher performance, with a PSNR of 38.14 dB compared to 35.23 dB for HIR-Diff.

- Task Complexity: The tasks are also distinct: HIR-Diff focuses on reconstructing multispectral images from N bands to N bands, whereas our work reconstructs N bands from a single compressed image (1 band to N bands), which presents additional challenges due to the low sampling rate. This makes our task more difficult and computationally demanding.

- Flexibility and Performance: Our method offers a more flexible and efficient framework, significantly outperforming HIR-Diff in both performance (PSNR: 38.14 dB vs. 35.23 dB) and computational efficiency. Although we incorporated the HIR-Diff approach into the second stage of our framework, the results clearly show that our approach, leveraging spectral embedding, fine-tuning, and guidance, outperforms existing methods for SCI tasks.

## A.6 LIMITATION ANALYSIS AND GENERALIZATION ACROSS DIVERSE DATASETS

### A.6.1 CLARIFICATION ON PSNR AND PERFORMANCE ON KAIST DATASET

For the four test datasets, our PSR-SCI achieves the best PSNR and SSIM on three of them (ICVL, NTIRE, and Harvard), with the exception of the KAIST dataset. To understand this discrepancy, we carefully compared the four test datasets with the training dataset (CAVE).

Our analysis revealed that the wavelengths and scene objects in the KAIST dataset closely resemble those in the CAVE training dataset. In contrast, the ICVL, NTIRE, and Harvard datasets are notably

different from CAVE in terms of spectral and scene diversity. This suggests that non-diffusion-based methods, which achieve over 40dB PSNR on KAIST, perform well due to their strong fitting to the training dataset. However, this overfitting comes at a cost: their generalization capability decreases significantly, leading to sharp performance drops on datasets that differ from the training data, such as ICVL, NTIRE, and Harvard (as evidenced by a sharp PSNR drop of over 3 dB when applied to the ICVL dataset).

In contrast, our PSR-SCI, as a diffusion-based method, does not prioritize overfitting to the training dataset. Instead, it leverages its generative capabilities to tackle challenges that non-diffusion models struggle with, such as reconstructing high-frequency details across diverse scenes and performing well on real-world datasets. This difference in focus allows PSR-SCI to maintain strong performance on datasets with varied characteristics (KAIST: 38.14dB, ICVL: 37+dB ), demonstrating better generalization compared to non-diffusion methods.

On the other hand, while our PSR-SCI method achieves a PSNR of 38.14 dB on the KAIST dataset, which is slightly lower than some recent end-to-end networks that exceed 39 dB (e.g., PADUT Li et al. (2023), RDLUF-MixS2 Dong et al. (2023), and LADE-DUN Wu et al. (2025)), it's important to note that our method excels in other quantitative metrics. Specifically, on the KAIST dataset, our method outperforms these latest end-to-end methods in terms of MUSIQ, MANIQA, and CLIP-IQA scores (Table 10).

Table 10: Comparison between PSR-SCI and related works on the KAIST dataset.

| Method | Category | Reference | PSNR (dB) ↑ | SSIM ↑ | LPIPS ↓ | MUSIQ ↑ | MANIQA ↑ | CLIP-IQA ↑ |
|---|---|---|---|---|---|---|---|---|
| PADUT-12stg Li et al. (2023) | Unfolding | ICCV 2023 | 38.89 | 0.974 | 0.03953 | 40.55469 | 0.24266 | 0.33274 |
| RDLUF-MixS2-9stg Dong et al. (2023) | Unfolding | CVPR 2023 | 39.57 | 0.974 | - | - | - | - |
| LADE-DUN-10stg Wu et al. (2024) | Unfolding | ECCV 2024 | 40.16 | 0.980 | 0.03186 | 41.29688 | 0.24890 | 0.34466 |
| SPECAT Yao et al. (2024) | Transformer | CVPR 2024 | **40.37** | **0.986** | **0.02785** | 41.72187 | 0.24882 | 0.34294 |
| MST-L | Transformer | CVPR 2022 | 35.18 | 0.948 | 0.06906 | 37.06562 | 0.21179 | 0.31166 |
| DAUHST-3stg | Unfolding | NeurIPS 2022 | 37.21 | 0.959 | 0.05718 | 37.64531 | 0.21808 | 0.29596 |
| DAUHST-SP2 He et al. (2024) | Suspace prior | Information Fusion 2024 | 37.61 | 0.966 | - | - | - | - |
| DiffSCI | Diffusion | CVPR 2024 | 35.28 | 0.916 | 0.07421 | 39.64512 | 0.23598 | 0.33152 |
| **PSR-SCI-D** | Diffusion | Ours | 38.14 | 0.967 | 0.02844 | **42.72969** | **0.25279** | **0.35602** |

Moreover, on the ICVL, NTIRE, and Harvard datasets, our PSR-SCI achieves the best PSNR, SSIM, and MANIQA metrics compared to the latest end-to-end methods (as presented in Table 2 in the main paper, we reproduce it here in Table 11 for easy reference). This demonstrates our method's excellent generalization ability across various datasets.

Table 11: Comparison of PSNR, SSIM, and MANIQA metrics across several zero-shot datasets.

| Dataset | Metric | DAUHST-3stg (NeurIPS 2022) | MST-L (CVPR 2022) | DPU-9stg (CVPR 2024) | SSR-L (CVPR 2024) | LADE-10stg (ECCV 2024) | DiffSCI (CVPR 2024) | PSR-SCI-D (Ours) | PSR-SCI -DPU (Ours) | PSR-SCI -SSR (Ours) |
|---|---|---|---|---|---|---|---|---|---|---|
| ICVL | PSNR↑ | 34.64 | 34.03 | 36.56 | 36.25 | 35.89 | 33.02 | 37.03 | **37.25** | 37.14 |
| | SSIM↑ | 0.890 | 0.885 | 0.918 | 0.914 | 0.904 | 0.868 | 0.918 | **0.923** | 0.918 |
| | MANIQA↑ | 0.200 | 0.209 | 0.200 | 0.209 | 0.210 | 0.207 | **0.217** | 0.216 | 0.213 |
| NTIRE | PSNR↑ | 34.44 | 33.04 | 36.25 | 35.44 | 33.58 | 32.79 | 36.44 | **36.62** | 35.53 |
| | SSIM↑ | 0.927 | 0.914 | 0.945 | 0.942 | 0.923 | 0.903 | 0.953 | **0.955** | 0.948 |
| | MANIQA↑ | 0.214 | 0.210 | 0.226 | 0.230 | 0.221 | 0.205 | 0.233 | 0.238 | **0.240** |
| Harvard | PSNR↑ | 25.57 | 24.01 | 27.05 | 25.93 | 28.02 | 24.68 | 26.90 | 28.58 | **29.02** |
| | SSIM↑ | 0.622 | 0.594 | 0.650 | 0.597 | 0.739 | 0.602 | **0.776** | 0.764 | 0.728 |
| | MANIQA↑ | 0.187 | 0.204 | 0.197 | 0.195 | 0.198 | 0.174 | 0.205 | 0.239 | **0.247** |

Additionally, our method provides superior reconstruction results on real data. Our approach retains capabilities for generation and local inpainting tasks, which non-diffusion methods cannot achieve. Therefore, despite the slightly lower PSNR on the KAIST dataset, our method offers considerable advantages in perceptual quality, generalization, and additional functionalities, indicating a substantial improvement over existing techniques.

### A.6.2 LIMITATION ON MODEL PARAMETER COUNT.

While our PSR-SCI method demonstrates significantly faster inference times compared to similar diffusion-based methods like DiffSCI (8.9 seconds vs. 85 seconds for 50 steps), it inherits a large number of parameters from the pre-trained diffusion models. This reliance on pre-trained models is crucial for leveraging their superior generative capabilities, which are essential for capturing high-frequency details in spectral image reconstruction.

However, this also means that our model's overall parameter count is larger compared to end-to-end networks like MST, as shown in Table 12. While the MST model demonstrates a lower parameter count, even when scaled up, its performance and generalization ability remain limited. Specifically, increasing the MST model size from 2.018M to 39.052M parameters only yields a modest PSNR improvement of 0.58 dB, and further scaling it to 480.535M parameters leads to a PSNR drop to 32.98 dB due to overfitting. This underscores the limitations of non-diffusion-based models, where simply increasing the parameter count cannot ensure better reconstruction performance.

In contrast, our PSR-SCI method leverages large pre-trained diffusion models in a novel framework designed for spectral image reconstruction. This approach enables our method to capture high-frequency spectral details effectively while avoiding overfitting. Unlike MST, our method demonstrates consistent performance improvements at larger scales, achieving a PSNR of 38.14 dB and an SSIM of 0.967. These gains are attributable to the innovative integration of trainable spectral embeddings and a fine-tuned diffusion refinement module, rather than simply the parameter count.

Table 12: Comparison of reconstruction performance for different model sizes and methods.

| Model | Parameters | Test PSNR (dB) ↑ | Test SSIM ↑ | Training Time |
|---|---|---|---|---|
| MST-L | 2.018M | 35.18 | 0.948 | - |
| MST-exp1 | 39.052M | 35.76 | 0.957 | 8.76h |
| MST-exp2 | 480.535M | 32.98 | 0.921 | 95.21h |
| PSR-SCI-D (Ours) | 1312M | **38.14** | **0.967** | 18h |

In summary, while our method's larger parameter count is a limitation, it is a necessary trade-off to achieve high-quality reconstructions by utilizing the generative power of pre-trained diffusion models. This ensures that our PSR-SCI method can deliver superior performance and faster inference times compared to other diffusion-based methods, albeit with a higher parameter count than some end-to-end networks.

## A.7 ANALYSIS OF TRAINING DATASETS FOR DIFFUSION MODELS

To address the reviewer's concern regarding the availability and suitability of hyperspectral image (HSI) datasets for training diffusion models, we conducted a comparative study on the impact of different training datasets. While the recently proposed HSIGene Pang et al. (2024b) offers a synthetic solution to dataset limitations, our experiments reveal significant differences in its utility compared to real datasets.

Our results indicate that real datasets, such as CAVE and Harvard, significantly improve the performance of diffusion models in reconstructing high-frequency details, achieving up to a +6.18 dB PSNR improvement. In contrast, incorporating synthetic data generated by HSIGene leads to a slight reduction in performance, with a -0.83 dB drop in PSNR and lower SSIM scores, as shown in Table 13. This demonstrates that the quality and diversity of high-frequency details in real data play a critical role in effective training.

Table 13: Impact of training datasets on diffusion model performance.

| Training Datasets | PSNR (dB) ↑ | SSIM ↑ |
|---|---|---|
| Pretrained Diffusion Only | 32.49 | 0.878 |
| CAVE (Real Dataset) | 38.14 | 0.967 |
| CAVE + HSIGene (Generated) | 37.31 | 0.959 |
| CAVE + Additional Real Dataset (Same Amount as HSIGene) | **38.67** | **0.972** |

While HSIGene offers an alternative to alleviate dataset scarcity for simpler networks, it is less effective for diffusion models. The limitations of HSIGene stem from its reliance on a restricted training dataset, resulting in generated images that often lack the diverse and high-quality high-frequency information required for advanced diffusion model training. Our findings underscore the necessity of real, high-quality datasets for optimizing the recovery of fine spectral details.

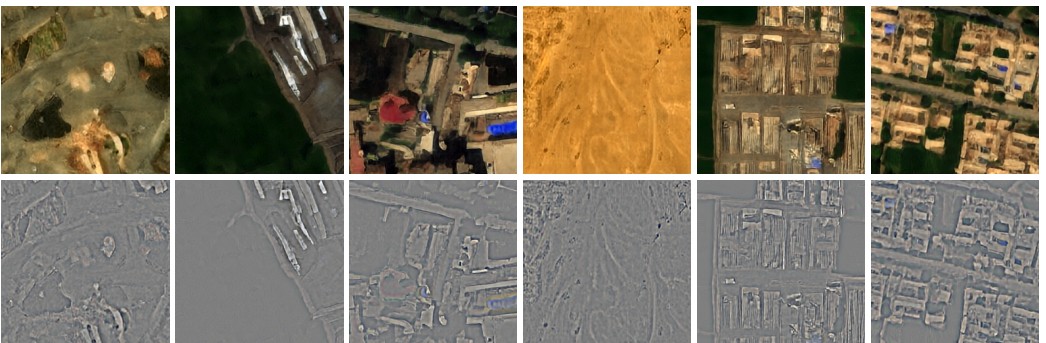

Figure 15: Visual comparison of raw and high-freq part of datasets generated by HSIGene Pang et al. (2024a)

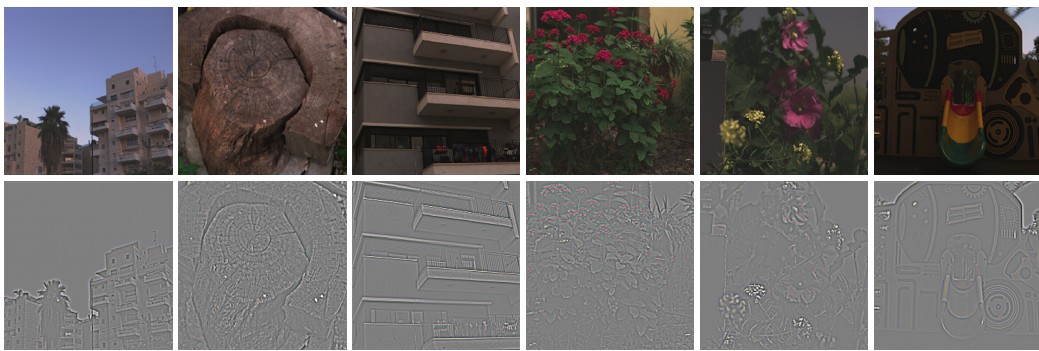

Figure 16: Visual comparison of raw and high-freq part of real captured datasets NTIRE2022 Arad et al. (2022)

We identified that a considerable proportion of high-frequency details in the hyperspectral images produced by HSIGene (Fig. 15) exhibit misalignments or imperfections (Fig. 16), which directly contradicts the goal of leveraging diffusion models to accurately restore high-frequency information. While HSIGene's synthetic images are beneficial for alleviating dataset limitations when training simpler networks, they prove less suitable for advanced models like diffusion. This limitation arises from HSIGene's reliance on a relatively restricted training dataset, which often results in generated images lacking the diversity and quality of high-frequency details necessary for effective diffusion model training.

Experiments incorporating additional real MSI datasets, such as Harvard, ICLV, and NTIRE2022, further highlight the importance of genuine data, demonstrating significant improvements in the capacity of diffusion models to recover detailed high-frequency features. These findings underscore that, while synthetic datasets like HSIGene can serve as a valuable augmentation resource for certain models, they fail to meet the stringent requirements of diffusion models designed for precise high-frequency reconstruction. Consequently, leveraging real, high-quality datasets is critical to fully exploit the capabilities of diffusion-based methods in hyperspectral imaging.

A.8    ADDITIONAL RESULTS ON SIMULATION AND REAL DATASETS

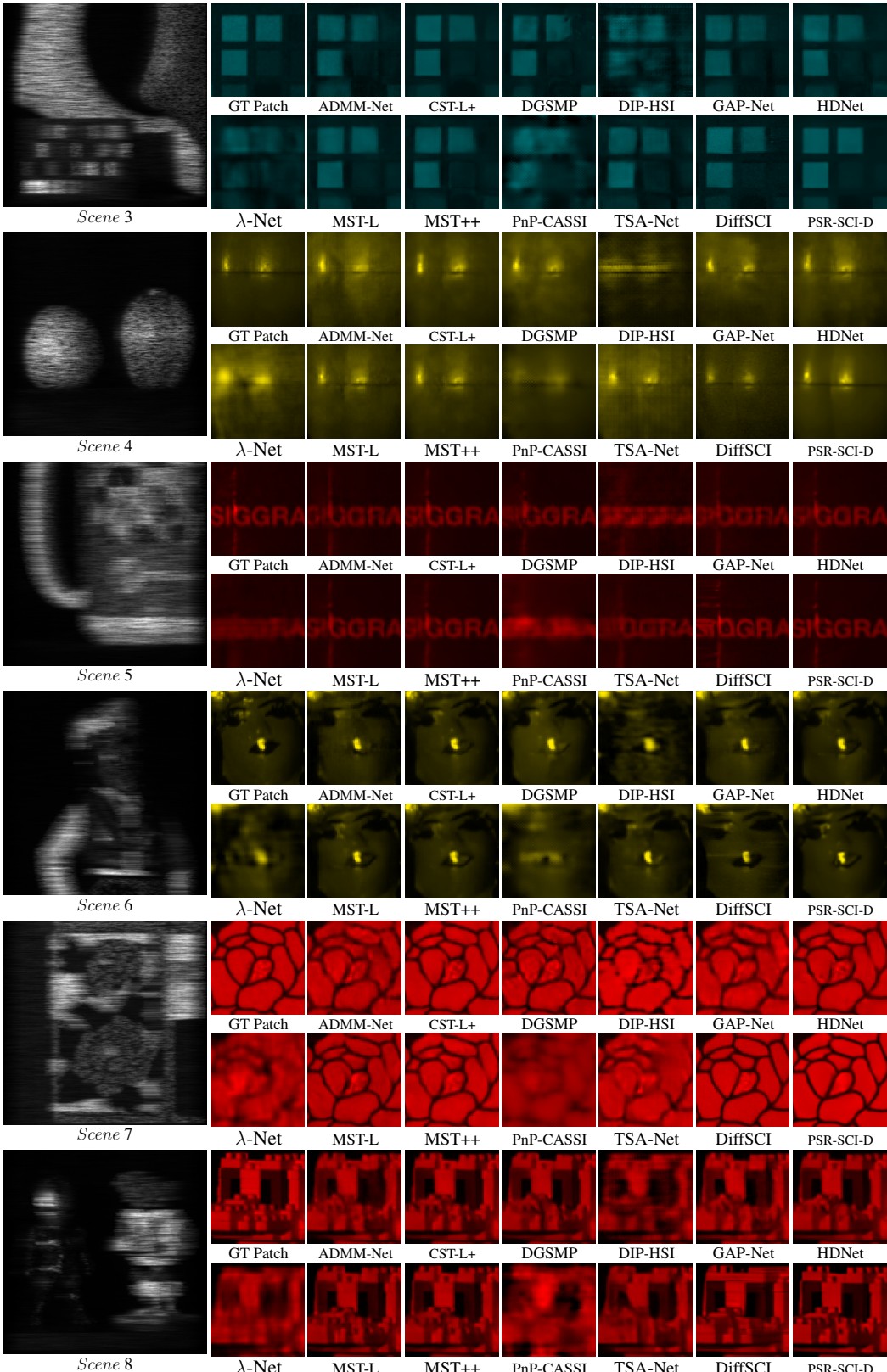

Figure 17: Visual comparison from *Scene* 3 to *Scene* 8 of the KAIST dataset.

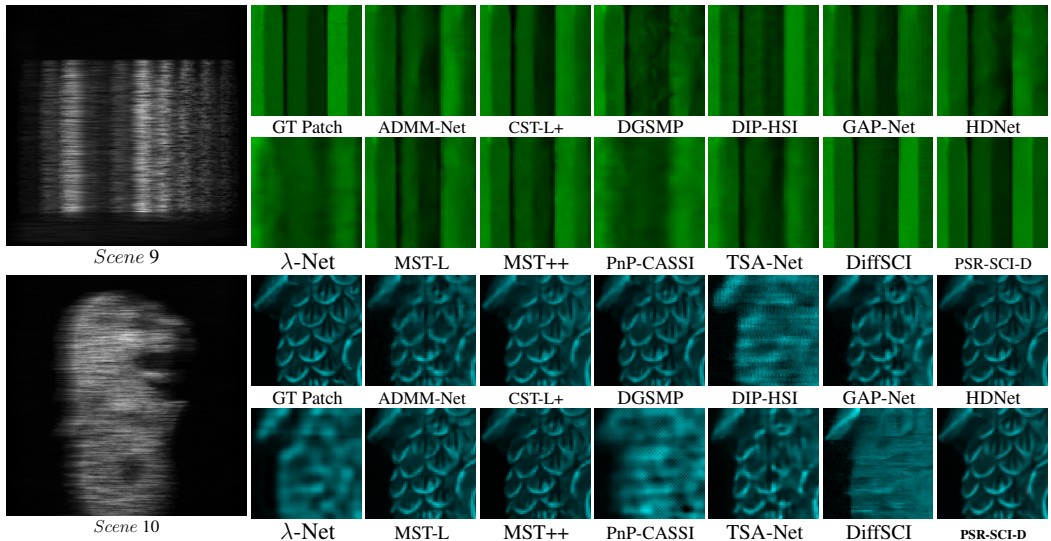

Figure 18: Visual comparison from *Scene* 9 to *Scene* 10 of the KAIST dataset.

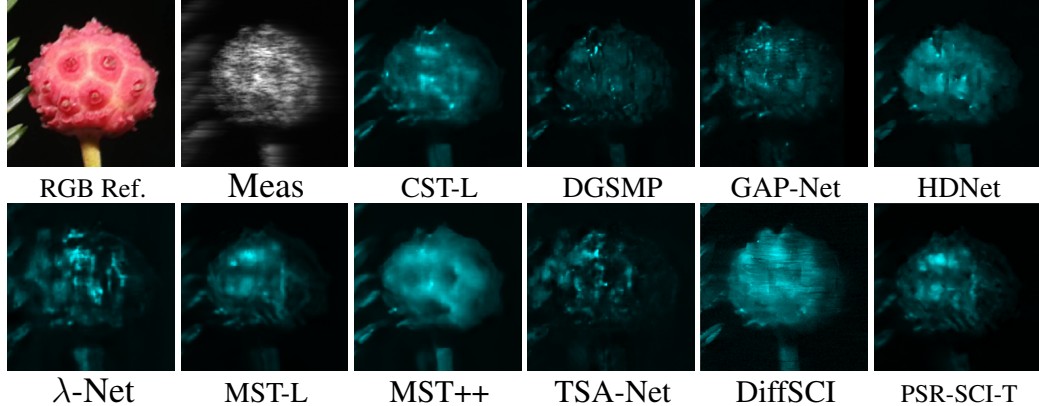

Figure 19: Visual comparison on *Scene* 2 of real dataset at wavelength 487nm.

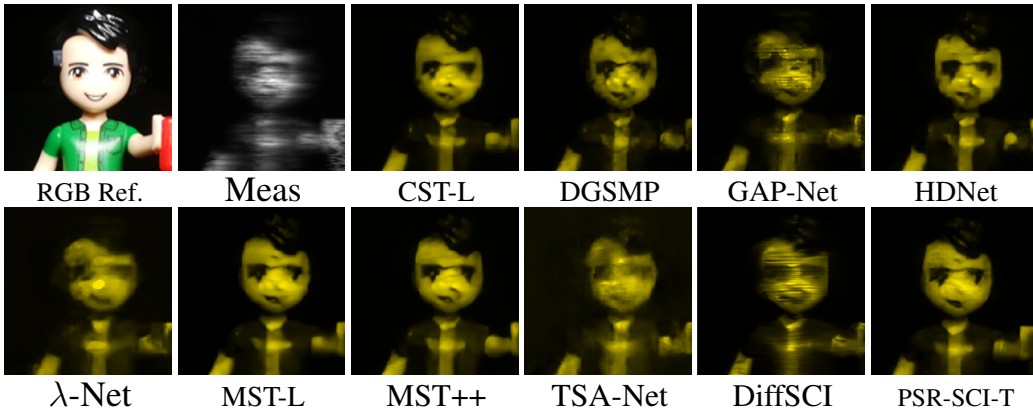

Figure 20: Visual comparison on *Scene* 3 of real dataset at wavelength 575.5nm.

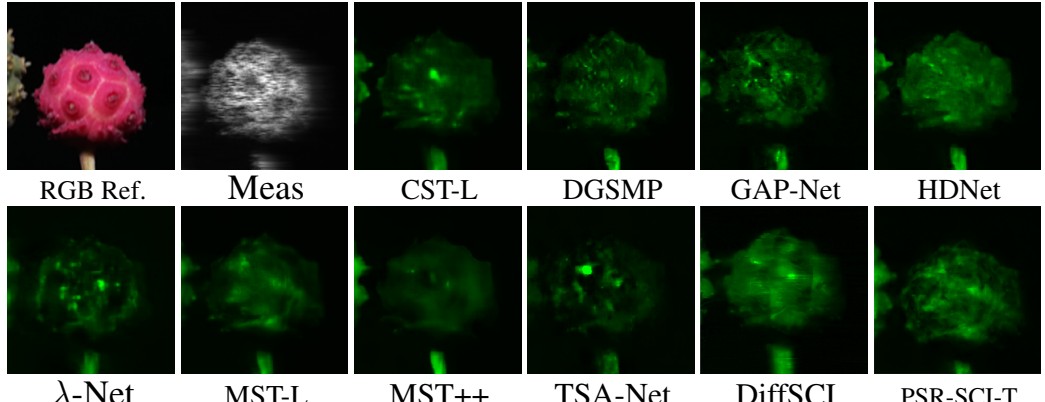

Figure 21: Visual comparison on $Scene$ 4 of real dataset at wavelength 536.5nm.

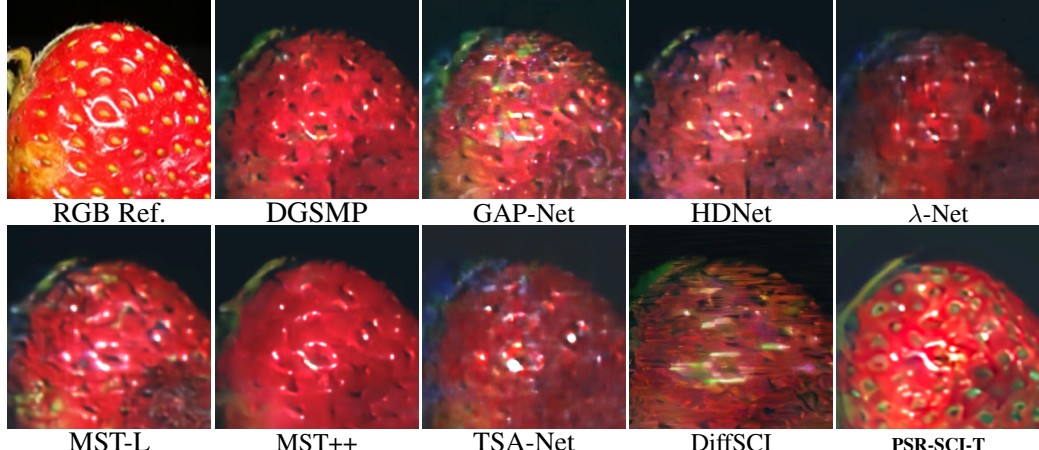

Figure 22: Visual comparison on real $Scene$ 5 in pseudo-RGB (604nm, 536.5nm and 481.5nm).

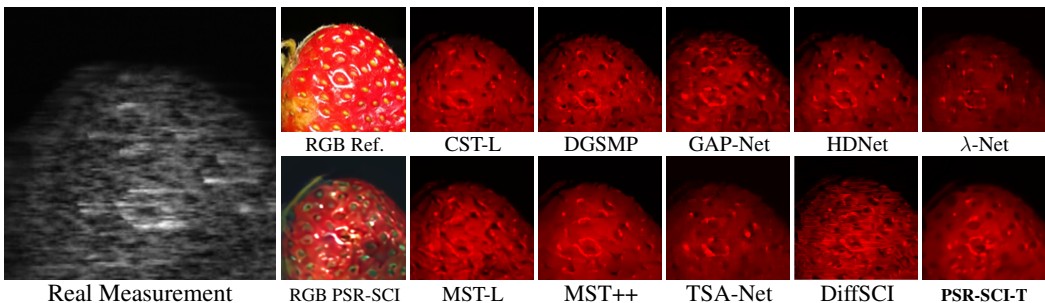

Figure 23: Visual comparison of SCI reconstruction models on real $Scene$ 5 at wavelength 648.0nm.

