# OpenReview forum: "Spectral Compressive Imaging via Unmixing-driven Subspace Diffusion Refinement"
_ICLR.cc/2025/Conference — ICLR 2025 Spotlight_

### Official Review · Reviewer_Z3XV · 2024-10-17

**Soundness:** 3
**Presentation:** 2
**Contribution:** 2
**Rating:** 6
**Confidence:** 5

**Summary:**

The paper introduces a novel framework called Predict-and-Unmixing-driven Subspace Refinement (PSR-SCI) for Spectral Compressive Imaging (SCI) reconstruction. SCI is an inherently ill-posed problem where traditional methods often struggle to recover high-frequency details. The proposed PSR-SCI framework starts with a cost-effective predictor that provides an initial estimate of the multispectral image (MSI). It then introduces an unmixing-driven reversible spectral embedding module to decompose the MSI into subspace images and spectral coefficients. This decomposition allows the adaptation of pre-trained RGB diffusion models, focusing refinement on high-frequency details efficiently, even with minimal MSI data. Additionally, the authors design a high-dimensional guidance mechanism with imaging consistency to enhance the model's effectiveness. The refined subspace image is reconstructed back into the MSI using the reversible embedding, yielding the final MSI with full spectral resolution. Experimental results on standard datasets like KAIST and zero-shot datasets such as NTIRE, ICVL, and Harvard demonstrate that PSR-SCI enhances visual quality and achieves PSNR and SSIM metrics comparable to some existing diffusion, transformer, and deep unfolding techniques.

**Strengths:**

1. The proposed high-dimensional guidance mechanism with imaging consistency is a noteworthy contribution that could enhance the model's performance.

2. The method is evaluated on both standard datasets and zero-shot datasets, showing competitive performance in PSNR and SSIM metrics compared to existing methods.

**Weaknesses:**

1. The motivation for using the unmixing-driven spectral embedding decomposition is not entirely clear. The paper does not sufficiently justify why this specific decomposition method is suitable for the SCI problem or how it leads to superior performance in terms of convergence speed, recovery accuracy, or error reduction. While the approach is innovative, its superiority over existing methods is not well-established.

2. The claim that there is limited training data available for MSIs compared to RGB images is questionable. Recent works have utilized large-scale MSI datasets to train diffusion models effectively (e.g., "HSIGene: A Foundation Model For Hyperspectral Image Generation," https://arxiv.org/abs/2409.12470, https://github.com/LiPang/HSIGene). This undermines the premise that data scarcity necessitates their approach.

3. The proposed method appears to have a high number of parameters and computational demands. It is unclear whether the performance gains are due to the novel method or simply the result of using a much larger model compared to others. Methods like DAUHST and MST use models with millions of parameters, whereas the proposed method uses billions. This raises concerns about the practicality and efficiency of the approach, especially considering the longer inference times.

4. Despite the innovative approach, the performance of PSR-SCI on the KAIST dataset is reported as 38.14dB PSNR, which is lower than some recent works that achieve PSNR values exceeding 39dB (e.g., PADUT [ICCV 2023], RDLUF-MixS2 [CVPR 2023], and "Latent Diffusion Prior Enhanced Deep Unfolding for Snapshot Spectral Compressive Imaging" [ECCV 2024]). This suggests that the proposed method may not offer a significant improvement over existing techniques.

**Questions:**

1. Could the authors elaborate on why the unmixing-driven spectral embedding decomposition is particularly suitable for SCI reconstruction? How does it contribute to faster convergence, higher recovery accuracy, or error reduction compared to other methods?

2. What is the total number of parameters in the proposed model, and how does it compare to other methods? Could the authors provide a detailed analysis of the computational complexity and inference times? How practical is the method for real-world applications given its computational demands?

3. Given that recent works have successfully trained diffusion models on large-scale MSI datasets, how do the authors address the claim of MSI data scarcity? Does this affect the necessity or uniqueness of the proposed approach?

4. To what extent does the high performance of PSR-SCI stem from the large number of parameters and depth of the model, as opposed to the proposed methodology itself? Have the authors conducted experiments to isolate these factors?

---

> ### Author Response · Authors · 2024-11-24
> **Response to Reviewer Z3XV**
>
> We thank the reviewer for recognizing the innovation of our approach, particularly the high-dimensional guidance mechanism with imaging consistency, as well as acknowledging our competitive performance on both standard and zero-shot datasets in terms of PSNR and SSIM, highlighting the practical value and effectiveness of our contributions. Please see below for our point-by-point responses to your comments.
>
> - **Comment**: The motivation for using the unmixing-driven spectral embedding decomposition is not entirely clear. The paper does not sufficiently justify why this specific decomposition method is suitable for the SCI problem or how it leads to superior performance in terms of convergence speed, recovery accuracy, or error reduction. While the approach is innovative, its superiority over existing methods is not well-established.
>
> > **@** The original intent behind designing the reversible unmixing module was to achieve dimensionality reduction of spectral information channels while ensuring accurate representation of spectral data and compatibility with the RGB space. Thus, we considered using an SVD-based approach for decomposition from the beginning. Our comparison results on KAIST dataset show that, **compared to using the SVD method as described in the HIR-Diff paper, our learnable unmixing module achieves superior reconstruction performance and inference speed.**
> >
> >| Method                         | Scene 1      | Scene 2      | Scene 3      | Scene 4      | Scene 5      | Scene 6      | Scene 7      | Scene 8      | Scene 9      | Scene 10     | Average              |
> >| ------------------------------ | ------------ | ------------ | ------------ | ------------ | ------------ | ------------ | ------------ | ------------ | ------------ | ------------ | -------------------- |
> >| **SVD**-band-select (HIR-Diff [1]) | 40.65 0.9810 | 36.25 0.9667 | 39.20 0.9795 | 51.10 0.9948 | 38.70 0.9909 | 40.60 0.9873 | 43.12 0.9862 | 40.80 0.9867 | 33.00 0.9623 | 46.60 0.9913 | 41.00 0.9827         |
> >| **URSe** (Ours)                | 50.54 0.9982 | 54.03 0.9990 | 49.61 0.9939 | 57.02 0.9994 | 49.28 0.9984 | 49.99 0.9984 | 47.75 0.9948 | 49.05 0.9977 | 50.57 0.9967 | 50.97 0.9988 | **50.88** **0.9975** |
> >
> >    Specifically, SVD-based methods generally exhibit inferior reconstruction quality compared to our designed module. The reason for this is that SVD models the relationships between spectral bands as simple linear transformations, whereas, in reality, most spectral relationships are inherently nonlinear. Although SVD provides an option of spectral decomposition, its speed advantage diminishes in multispectral image. Conversely, our learnable module not only maintains competitive inference speed but also achieves superior compression and reconstruction performance.
> >
> >    **Further investigation of URSe for spectral embedding** (Our URSe shows robustness with noise)
> >
> >    To evaluate the robustness of our spectral embedding module **URSe** within the diffusion model, we conducted additional ablation experiments detailed in the table below. We introduced Gaussian perturbations to **E** and found that our method remains robust to zero-mean Gaussian noise with variances from **0** to **0.1**, maintaining a PSNR exceeding **37 dB** even under significant noise (variance **0.1**). In contrast, the SVD-based method is sensitive to noise; with light noise of variance **0.01**, its PSNR drops by **0.72 dB**, and with noise of variance **0.1**, it drops by **10.05 dB**. These results highlight the superior noise robustness of our URSe module compared to SVD-based approaches.
> >
> >    Table. Ablation study on the robustness of spectral embedding for the diffusion model on KAIST dataset.
> >
> >| Type                          | Noise (Gaussian) perturbation | PSNR      | SSIM       |
> >| ----------------------------- | ------------------------------------------ | --------- | ---------- |
> >| URSe  (Ours)                    | 0                                          | **38.14** | **0.9670** |
> >|                               | 0.01                                       | 38.1      | 0.9665     |
> >|                               | 0.1                                        | 37.04     | 0.9567     |
> >| SVD-band-select (from HIR-diff [1]) | 0                                          | 36.87     | 0.9628     |
> >|                               | 0.01                                       | 36.15     | 0.9623     |
> >|                               | 0.1                                        | 26.82     | 0.9383     |
>
> [1] Pang, Li, et al. "HIR-Diff: Unsupervised Hyperspectral Image Restoration Via Improved Diffusion Models." Proceedings of the IEEE/CVF Conference on Computer Vision and Pattern Recognition. 2024.

---

> > ### Author Response · Authors · 2024-11-24
> > **Response to Reviewer Z3XV (continued)**
> >
> > >**For comment-1: The motivation for using the unmixing-driven spectral embedding: overall SCI comparion with SVD-ban-select based method HIR-Diff** (Our PSR-SCI achieves **38.14 dB**, compared to **35.23 dB** for HIR-Diff.)
> > >
> > >**@**: Although [1] was not originally designed for the Snapshot Compressive Imaging (SCI) task we address in this paper, for a more comprehensive comparison, we incorporated it into the second stage of our two-stage framework (while keeping the first stage unchanged). We compared HIR-Diff with our proposed PSR-SCI, and the reconstruction performance for snapshot tasks is summarized below. The results indicate that HIR-Diff achieves reasonable reconstruction quality, but its performance metrics lag behind state-of-the-art methods. Our approach, leveraging guidance and additional trainable modules, demonstrates clear advantages over HIR-Diff, in terms of performance metrics.
> > >
> > >   | Method                                                     | Reference | PSNR  | SSIM  |
> > >    | ---------------------------------------------------------- | --------- | ----- | ----- |
> > >   | HIR-DIff (+our fisrt stage-to ensure it works for SCI) [1] | CVPR-2024 | 35.23 | 0.959 |
> > >    | PSR-SCI-D                                                  | Ours      | 38.14 | 0.967 |
> > >
> >
> >
> >
> > - **Comment**: The claim that there is limited training data available for MSIs compared to RGB images is questionable. Recent works have utilized large-scale MSI datasets to train diffusion models effectively (e.g., "HSIGene: A Foundation Model For Hyperspectral Image Generation," https://arxiv.org/abs/2409.12470, https://github.com/LiPang/HSIGene). This undermines the premise that data scarcity necessitates their approach.
> >
> > >
> > >    **@**: We trained the diffusion model using real MSI datasets such as Harvard and compared this to using datasets generated by HSIGene (It is also worth noting that HSIGene was released in September 2024, while the deadline for ICLR is also in September). **The results below demonstrate that real collected datasets significantly enhance the diffusion model's ability to recover high-frequency details (+6.18dB PSNR), while datasets generated by HSIGene somewhat reduce the model's performance in terms of PSNR (-0.83dB) and SSIM.**
> > >
> > >    Table 1: Added ablation study on the used datasets for fine-tuning diffusion model
> > >
> > >    | Training datasets                                           | PSNR Aver | SSIM Aver |
> > >    | ----------------------------------------------------------- | --------- | --------- |
> > >    | only pretrained diffusion                                   | 32.49     | 0.878     |
> > >    | CAVE (real dataset)                                         | 38.14     | 0.967     |
> > >    | CAVE + HSIGene (generated）                                 | 37.31     | 0.959     |
> > >    | CAVE + additonal real dataset（the same amount as HSIGene） | **38.67** | **0.972** |
> > >
> > >    By analyzing detailed examples of hyperspectral images generated by HSIGene (see revised appendix for details), we found that a relatively large proportion of high-frequency details are misaligned or contain flaws, which contradicts our original goal of restoring high-frequency details via diffusion models. HSIGene-generated hyperspectral images are suitable for addressing the lack of datasets when training simple networks but are not suitable for training models like diffusion, because it is itself a trained diffusion model with limit datasets.
> >
> > [1] Pang, Li, et al. "HIR-Diff: Unsupervised Hyperspectral Image Restoration Via Improved Diffusion Models." Proceedings of the IEEE/CVF Conference on Computer Vision and Pattern Recognition. 2024.

---

> > > ### Author Response · Authors · 2024-11-24
> > > **Response to Reviewer Z3XV (continued)**
> > >
> > > - **Comment**: The proposed method appears to have a high number of parameters and computational demands. It is unclear whether the performance gains are due to the novel method or simply the result of using a much larger model compared to others. Methods like DAUHST and MST use models with millions of parameters, whereas the proposed method uses billions. This raises concerns about the practicality and efficiency of the approach, especially considering the longer inference times.
> > >
> > > > **@**: To address your concerns, we conducted additional experiments by increasing the model size of the transformer-based MST. We observed that increasing the model size from **2.018M** to **39.052M** parameters provided only a **0.58 dB** improvement in PSNR. However, further increasing the size to **480.535M** resulted in a PSNR drop to **32.98 dB** due to overfitting, which harmed testing performance. This indicates that merely enlarging model size doesn't guarantee better results and can be detrimental.
> > > >
> > > > In contrast, our diffusion-based PSR-SCI maintains advantages at larger scales without overfitting. Notably, like most stable or latent diffusion methods, the majority of our parameters are inherited from pre-trained diffusion models. While these models are large, they are essential for generating realistic high-frequency details that smaller models can't capture effectively.
> > > >
> > > > Regarding efficiency, our method demonstrates significantly faster inference times compared to existing diffusion models for SCI. Specifically, our PSR-SCI requires **12.9 s**, whereas DiffSCI (CVPR 2024) takes **130.15 s**. This substantial reduction highlights the practicality of our approach despite the larger model size.
> > > >
> > > >  In summary, our performance gains stem from our novel method effectively leveraging pre-trained diffusion models, not just from increased model size, achieving superior reconstruction quality with practical efficiency.
> > > >
> > > > | Model     | Parameters | Test PSNR | Test SSIM | Training Time |
> > > > | --------- | ---------- | --------- | --------- | ------------- |
> > > > | MST-L     | 2.018M     | 35.18     | 0.948     | -             |
> > > > | MST-exp1  | 39.052M    | 35.76     | 0.957      | 8.76h         |
> > > > | MST-exp2  | 480.535M   | 32.98     | 0.921     | 95.21h        |
> > > > | PSR-SCI-D | 1312M     | **38.14** | **0.967** | 18h           |
> > >
> > > - **Comment**: Could the authors provide a detailed analysis of the computational complexity and inference times?
> > >
> > > >    **@**: We appreciate your concern regarding the computational cost and inference time. To address this, we have prepared a table that details the  **inference time** for our proposed method compared to other diffusion-based algorithm:
> > > >
> > > >    | Method    | Category  | Reference | Inference Time (50 steps, Best for PSR-SCI) | Inference Time (600 steps, Best for DiffSCI) |
> > > >    | --------- | --------- | --------- | ------------------------------------------- | -------------------------------------------- |
> > > >    | DiffSCI   | Diffusion | CVPR 2024 | 130.15s                                     | 865.81s                                      |
> > > >    | PSR-SCI-D | Diffusion | Ours      | **12.90s**                                      | **74.76s**                                       |
> > > >
> > > >    The observation is that our PSR-SCI is 10$\times$ faster than DffSCI (CVPR 2024). Our design of the **URSe module** and the two-stage pipeline enables our method to achieve superior inference speed and reduced computational load compared to other diffusion-based approaches. The initial predictor in our PSR-SCI framework is lightweight, and the proposed spectral embedding is both small and efficient. Similar to most stable diffusion or latent diffusion-based methods, the majority of our parameters are inherited from the pre-trained diffusion models we utilize.

---

> > > > ### Author Response · Authors · 2024-11-24
> > > > **Response to Reviewer Z3XV (continued)**
> > > >
> > > > - **Comment**: Despite the innovative approach, the performance of PSR-SCI on the KAIST dataset is reported as 38.14dB PSNR, which is lower than some recent works that achieve PSNR values exceeding 39dB (e.g., PADUT [ICCV 2023], RDLUF-MixS2 [CVPR 2023], and "Latent Diffusion Prior Enhanced Deep Unfolding for Snapshot Spectral Compressive Imaging" [ECCV 2024]). This suggests that the proposed method may not offer a significant improvement over existing techniques.
> > > >
> > > > > **@**: To address your concerns, we have conducted additional comparative experiments, as shown below. While our PSR-SCI method achieves a PSNR of **38.14 dB** on the KAIST dataset, which is slightly lower than some recent end-to-end networks that exceed **39 dB** (e.g., PADUT [ICCV 2023], RDLUF-MixS2 [CVPR 2023], and LADE-DUN [ECCV 2024]), it's important to note that our method excels in other quantitative metrics. Specifically, on the KAIST dataset, our method outperforms these latest end-to-end methods in terms of **MUSIQ**, **MANIQA**, and **CLIP-IQA** scores.
> > > > >
> > > > > Moreover, on the **ICVL**, **NTIRE**, and **Harvard** datasets, our PSR-SCI achieves the best **PSNR**, **SSIM**, and **MANIQA** metrics compared to the latest end-to-end methods, including the LADE-DUN you mentioned. This demonstrates our method's excellent generalization ability across various datasets.
> > > > >
> > > > > Additionally, our method provides superior reconstruction results on real data. Our approach retains capabilities for generation and local inpainting tasks, which non-diffusion methods cannot achieve. Therefore, despite the slightly lower PSNR on the KAIST dataset, our method offers considerable advantages in perceptual quality, generalization, and additional functionalities, indicating a substantial improvement over existing techniques.
> > > > >
> > > > > | Algorithms | Category    | Reference | PSNR↑     | MUSIQ↑    | MANIQA↑    | CLIP-IQA↑  |
> > > > > | ---------- | ----------- | --------- | --------- | --------- | ---------- | ---------- |
> > > > > | DPU-9stg   | Unfolding   | CVPR 2024 | 40.52     | 40.99     | 0.2487     | 0.3277     |
> > > > > | SSR-L      | Unfolding   | CVPR 2024 | **40.69** | 41.73     | 0.2504     | 0.3415     |
> > > > > | MST-L      | Transformer | CVPR 2022 | 35.18     | 37.06     | 0.2117     | 0.3116     |
> > > > > | DiffSCI    | Diffusion   | CVPR 2024 | 35.28     | 39.64     | 0.2359     | 0.3315     |
> > > > > | PSR-SCI-D  | Diffusion   | Ours      | 38.14     | **42.73** | **0.2527** | **0.3560** |
> > > > >
> > > > > | Dataset | Metric  | DAUHST-3stg (NeurIPS 2022) | MST-L (CVPR 2022) | DPU-9stg (CVPR 2024) | SSR-L (CVPR 2024) | DiffSCI (CVPR 2024) | Latent Diffusion Prior (ECCV 2024) | PSR-SCI-D (Ours) |
> > > > > | ------- | ------- | -------------------------- | ----------------- | -------------------- | ----------------- | ------------------- | ---------------------------------- | ---------------- |
> > > > > | ICVL    | PSNR↑   | 34.64                      | 34.03             | 36.56                | 36.25             | 33.02               | 35.89                              | **37.03**        |
> > > > > |         | SSIM↑   | 0.890                      | 0.885             | **0.918**            | 0.914             | 0.868               | 0.904                              | **0.918**        |
> > > > > |         | MANIQA↑ | 0.200                      | 0.209             | 0.200                | 0.209             | 0.207               | 0.210                              | **0.217**        |
> > > > > | NTIRE   | PSNR↑   | 34.44                      | 33.04             | 36.25                | 35.44             | 32.79               | 33.58                              | **36.44**        |
> > > > > |         | SSIM↑   | 0.927                      | 0.914             | 0.945                | 0.942             | 0.903               | 0.923                              | **0.953**        |
> > > > > |         | MANIQA↑ | 0.214                      | 0.210             | 0.226                | 0.230             | 0.205               | 0.221                              | **0.233**        |
> > > > > | Harvard | PSNR↑   | 25.57                      | 24.01             | 27.05                | 25.93             | 24.68               | **28.02**                          | 26.90            |
> > > > > |         | SSIM↑   | 0.622                      | 0.594             | 0.650                | 0.597             | 0.602               | 0.739                              | **0.776**        |
> > > > > |         | MANIQA↑ | 0.187                      | 0.204             | 0.197                | 0.195             | 0.174               | 0.198                              | **0.205**        |

---

> > > > > ### Comment · Reviewer_Z3XV · 2024-11-25
> > > > >
> > > > > Thank you for your detailed responses and the effort to address my concerns. However, key issues remain unresolved, and the current quality of the paper does not meet the standards of a top-tier conference like ICLR, especially when compared to other submissions.
> > > > >
> > > > > 1. Despite providing additional experimental results comparing your URSe module to SVD methods, you have not clearly explained how your spectral embedding fundamentally improves SCI reconstruction. A deeper theoretical justification is necessary to demonstrate why your approach offers advantages in convergence speed, reconstruction accuracy, or error reduction specifically for SCI tasks.
> > > > >
> > > > > 2. Using perceptual metrics like LPIPS, CLIPIQA, MUSIQ, and MANIQA for evaluating spectral images is inappropriate. These metrics are designed for natural RGB images and do not apply to spectral data, which is primarily used in scientific applications where fidelity and quantitative accuracy are crucial. Transforming spectral images to RGB for such evaluations leads to loss of important spectral information. Emphasizing these metrics does not effectively demonstrate the superiority of your method in SCI reconstruction.
> > > > >
> > > > > 3. Your method's PSNR on the KAIST dataset remains below that of recent works. In the SCI field, metrics like PSNR and SSIM that assess reconstruction fidelity are more pertinent than perceptual metrics. You should acknowledge this limitation in your paper and provide insights into how your method contributes despite lagging in these key metrics.
> > > > >
> > > > > 4. Some of your responses and explanations divert from the main concerns and attempt to overwhelm and distract the reviewer, which is counterproductive. Please focus on directly addressing these issues with clear and concise explanations to strengthen your work.
> > > > >
> > > > > I am open to further discussion and willing to reconsider my evaluation if these concerns are adequately addressed. If there are limitations in your work, it is better to acknowledge them directly. Nevertheless, the inferior performance in PSNR and SSIM remains a major concern.

---

> ### Author Response · Authors · 2024-11-25
> **Response to Reviewer Z3XV in terms of PSNR**
>
> > **Comment**: Nevertheless, the inferior performance in PSNR and SSIM remains a major concern.
>
> >**@**: Thank you for your constructive feedback. We agree that acknowledging limitations is an important aspect of presenting scientific work transparently, it is indeed an area where improvement is always welcome.
>
> >In addition, we would like to respectfully point out that **although our PSR-SCI achieves suboptimal PSNR on the KAIST dataset, our model was tested on four datasets, and on three of them—the ICVL, NTIRE, and Harvard datasets—it achieved the best PSNR and SSIM results**. Additionally, our method provides superior reconstruction results on real data, retains capabilities for generation and local inpainting tasks, which non-diffusion methods cannot achieve. Therefore, while we acknowledge that there is room for improvement in some aspects, we respectfully disagree with the comment that "inferior performance in PSNR and SSIM remains a major concern." We will, however, make sure to present these clarifications and the strengths of our method more explicitly in the revised paper. Thank you again for your thoughtful comments.

---

> > ### Comment · Reviewer_Z3XV · 2024-11-26
> >
> > Thank you for your detailed responses and the additional experimental results. While I appreciate the efforts you've made to address my concerns, several critical issues remain unresolved, which significantly affect my overall assessment of your work.
> >
> > 1. Your empirical comparisons with SVD-based methods show that your learnable unmixing module achieves better reconstruction performance and inference speed. However, the theoretical foundation explaining how your spectral embedding fundamentally improves SCI reconstruction remains insufficiently addressed. A deeper analysis is necessary to elucidate how your approach offers specific advantages for SCI tasks. Simply attributing the improvements to the nonlinear modeling of spectral relationships does not provide a fundamental understanding.
> >
> > 2. I previously mentioned that using perceptual metrics such as LPIPS, CLIPIQA, MUSIQ, and MANIQA in RGB space is inappropriate for evaluating spectral images. It is concerning that you did not directly address this point in your response or the updated paper. Spectral images are primarily used in scientific applications where spectral fidelity and quantitative accuracy are crucial. Evaluating them using metrics designed for natural RGB images, even after conversion, does not adequately reflect their reconstruction quality. I expected a direct response to this concern, and the lack of one is disappointing.
> >
> > 3. Your method's PSNR on the KAIST dataset remains below that of recent works by a significant margin (more than 2 dB), which raises concerns about its competitiveness in key aspects of SCI reconstruction. The KAIST dataset is a widely recognized benchmark in the SCI community, and performance on this dataset is crucial. While I acknowledge that your method achieves the best PSNR and SSIM results on the ICVL, NTIRE, and Harvard datasets, it is essential to address the performance gap on the KAIST dataset. This discrepancy needs to be thoroughly analyzed and discussed in the paper.
> >
> > 4. Although you mention that your method achieves faster inference times compared to other diffusion-based approaches, deploying a model requiring tens of seconds is impractical. As a researcher, I am not convinced to use such a complex and resource-intensive algorithm when more efficient models like MST are available. Furthermore, given that your method is slower and performs worse on the KAIST dataset, its practical utility is questionable.
> >
> > After reading the newest manuscript multiple times, I find that these critical issues have not been adequately addressed. Providing a stronger justification for your spectral embedding method, using appropriate evaluation metrics for spectral images, thoroughly discussing performance discrepancies on standard benchmarks, and offering a detailed analysis of computational practicality are very important. Currently, the paper and your responses do not meet the standards of ICLR. I remain open to further discussion and look forward to the authors addressing these concerns.

---

> > > ### Author Response · Authors · 2024-11-26
> > > **Response to Reviewer Z3XV (Clarifying PSNR Performance and the Generalization Strength of PSR-SCI Across Diverse Datasets))**
> > >
> > > Dear Reviewer Z3XV,
> > >
> > > Thank you for your valuable feedback. We truly appreciate your insightful comments, especially regarding the use of perceptual metrics such as **LIPIQA, MUSIQ, and MANIQA**. You are absolutely correct that these metrics operate in the RGB space, creating a gap when applied to the spectral domain. **The primary motivation for using these metrics is their prevalence in diffusion-based methods**, even though they are not perfect for evaluating spectral reconstruction. In spectral domain, we use them as supplementary metrics **to measure the spatial accuracy of reconstruction results, alongside PSNR and SSIM, which remain our primary evaluation metrics.**
> > >
> > > Regarding your observation on PSNR and perceptual metrics, we initially refrained from responding directly because we were conducting further experiments to thoroughly analyze this point. We are pleased to share the conclusion of these experiments now.
> > >
> > > For the four test datasets, our PSR-SCI achieves the best PSNR and SSIM on three of them (ICVL, NTIRE, and Harvard), with the exception of the KAIST dataset. To understand this discrepancy, we carefully compared the four test datasets with the training dataset (CAVE).
> > >
> > > **Our analysis revealed that the wavelengths and scene objects in the KAIST dataset closely resemble those in the CAVE training dataset. In contrast, the ICVL, NTIRE, and Harvard datasets are notably different from CAVE in terms of spectral and scene diversity**. This suggests that non-diffusion-based methods, which achieve over 40dB PSNR on KAIST, perform well due to their strong fitting to the training dataset. However, this overfitting comes at a cost: their generalization capability decreases significantly, leading to sharp performance drops on datasets that differ from the training data, such as ICVL, NTIRE, and Harvard (as evidenced by a sharp PSNR drop of over 3 dB when applied to the ICVL dataset).
> > >
> > > In contrast, our PSR-SCI, as a diffusion-based method, does not prioritize overfitting to the training dataset. Instead, it leverages its generative capabilities to tackle challenges that non-diffusion models struggle with, such as reconstructing high-frequency details across diverse scenes and performing well on real-world datasets. This difference in focus allows PSR-SCI to maintain strong performance on datasets with varied characteristics (**KAIST: 38.14dB, ICVL: 37+dB** ), demonstrating better generalization compared to non-diffusion methods.
> > >
> > > We hope this explanation addresses your concerns and provides clarity on the broader strengths of PSR-SCI. Thank you again for your insightful comments, which have helped us better articulate the contributions and capabilities of our method.
> > >
> > >
> > > thank you,
> > >
> > > the authors

---

> ### Author Response · Authors · 2024-11-26
> **Response to Reviewer Z3XV (detailed investigation of unmixing-driven spectral embedding)**
>
> Dear Reviewer Z3XV,
>
> Thank you for raising an important point about the theoretical foundation of our spectral embedding and its advantages for SCI reconstruction. After thorough experimental verification, we appreciate the opportunity to share what we found and elaborate on how it may address your concerns.
>
> ### 1. Limitations of SVD-Based and Non-Learnable Approaches
> SVD-based and band-selection methods, as shown in HIR-Diff and similar works, offer a linear and non-trainable means of spectral decomposition. However, such methods are inherently limited in two key ways:
> - **Forward-Decomposition Error Amplification:** These approaches cannot adaptively fine-tune the reverse process, leading to amplified errors introduced during the forward decomposition when reconstructing the spectral image.
> - **Misalignment with Spectral Characteristics:** SVD-based methods primarily focus on rough spectral-to-RGB mapping, which fails to capture the complex nonlinear relationships between spectral bands. This mismatch results in deviations between the diffusion model's outputs and the true spectral images due to differences in wavelength ranges (RGB: 460–650 nm vs. spectral imaging: 750–2500 nm).
>
> ### 2. Key Advantages of our URSe for SCI
> Our learnable URSe module is specifically designed to overcome these limitations through the following innovations:
> - **Fine-Tunable Spectral Embedding:** Unlike static SVD-based methods, URSe is fine-tuned on spectral image datasets, enabling it to learn task-specific nonlinear relationships and accurately align the spectral domain with pre-trained RGB diffusion models. This ensures that the model’s outputs are not just approximate reconstructions but are tailored to the spectral imaging domain. In addition, URSe is reversible, enabling high-quality spectral guidance for diffusion models.
> - **Noise Robustness:** URSe is robust to noise, a crucial consideration in practical SCI scenarios where measurements often contain noise, as highlighted in literature such as DAUHST and MST. This robustness ensures consistent performance in real-world settings.
> - **Enhanced Convergence and Accuracy:** By providing a more accurate and adaptive spectral embedding, URSe accelerates the diffusion process and achieves better reconstructions. For example, on the KAIST dataset, our method reduces inference time from **312.43s** to **12.9s** and improves PSNR from **33.42dB** to **38.14dB** compared to applying diffusion directly without URSe.
>
> ### 3. Compatibility with SCI Reconstruction Goals
> The core contribution of URSe lies in its ability to bridge the gap between RGB diffusion models and spectral imaging tasks. By aligning the learned spectral embeddings with the spectral image space:
> - It ensures that the diffusion model generates results that better represent the full spectral range.
> - It mitigates domain mismatch issues between RGB-based models and spectral data, leveraging the pre-trained RGB model while adapting it to the unique characteristics of spectral imaging.
>
> **Summarize in short**: Our approach extends beyond simple nonlinear modeling of spectral relationships. The learnable nature of URSe allows it to adapt dynamically, minimizing forward-reconstruction errors and ensuring robust, noise-resilient performance. This is particularly critical for SCI tasks where both accuracy and computational efficiency are paramount.
>
> We appreciate your suggestion and we are updating our manuscript to include this detailed analysis to better explain the theoretical foundation and specific advantages of our approach for SCI reconstruction. Your feedback has helped us highlight this point, thank you!

---

> > ### Comment · Reviewer_Z3XV · 2024-11-26
> >
> > I appreciate your detailed responses, which have clarified several aspects of your work. However, after careful consideration, I remain of the opinion that the overall contribution of the paper is incremental and does not represent a significant advancement in the field. Additionally, it appears that the proposed clarifications have not been fully integrated into the manuscript. While I have no further technical concerns, the paper itself must reflect these improvements to ensure that readers can fully understand the positioning of your work within the community and its inherent limitations. Since ICLR permits paper revisions during the rebuttal period, I strongly encourage you to update the manuscript accordingly. I look forward to reviewing an updated version that appropriately incorporates these clarifications and addresses the issues highlighted.

---

> ### Author Response · Authors · 2024-11-27
> **Response to Reviewer Z3XV (the revised PDF has been updated)**
>
> Dear Reviewer Z3XV,
>
> Thank you for your continued discussion and for highlighting the need for clearer integration of our clarifications. We have carefully revised both the main manuscript and the appendix to fully incorporate the key clarifications you suggested (see the revised PDF).
>
> Specifically, we have enhanced the discussion of our contributions, provided more detailed explanations of our methodology, and thoroughly addressed the limitations of our work. Due to space constraints, we have included most of the detailed clarifications, including supplementary tables and figures, in the appendix. We appreciate your understanding regarding these necessary adjustments to fit within the page limits. For your convenience, the appendix includes:
>
> - Sec. A.1: Implementation Details
> - Sec. A.2: Hyper-parameters Optimization
> - Sec. A.3: Subspace Diffusion with High-dimensional Guidance
> - Sec. A.4: Explanation of the Unmixing-driven Spectral Embedding Approach
> - Sec. A.5: Framework Architecture Analysis and Performance Evaluation
> - Sec. A.6: Limitation Analysis and Generalization Across Diverse Datasets
> - Sec. A.7: Analysis of Training Datasets for Diffusion Models
> - Sec. A.8: Additional Experimental Results
>
>
> Thank you again for your feedback, which helped us to enhance our work!
>
> Best regards,
>
> the authors

---

> > ### Comment · Reviewer_Z3XV · 2024-11-28
> >
> > I have read the revised paper and appreciate the efforts made to incorporate the suggested improvements. The revisions have addressed the concerns I previously raised, and the updated explanations and additional analyses significantly enhance the clarity and completeness of the manuscript. I have no further comments at this time and have raised my score to "Borderline Accept". Thank you for your thoughtful responses and thorough updates.

---

> ### Author Response · Authors · 2024-11-28
> **Response to Reviewer Z3XV**
>
> Dear Reviewer Z3XV,
>
> We truly appreciate the time you took to review our revised manuscript and for raising your score.
>
> It means a lot to us that the revisions addressed your concerns and that the updates enhanced the clarity and completeness of the work. We’re really thankful for your contribution to making our paper better.
>
> Thank you again for your support and for recognizing our efforts!
>
> Best regards,
>
> the authors

---

### Official Review · Reviewer_HqgD · 2024-10-27

**Soundness:** 3
**Presentation:** 3
**Contribution:** 3
**Rating:** 8
**Confidence:** 5

**Summary:**

The proposed Predict-and-Unmixing-driven Subspace-Refine framework (PSR-SCI) addresses the challenges in Spectral Compressive Imaging (SCI) reconstruction, particularly the difficulty of recovering high-frequency details from ill-posed inverse problems. By leveraging a cost-effective prediction module, unmixing-driven reversible spectral embedding, and pre-trained RGB diffusion models, PSR-SCI demonstrates superior visual quality and competitive performance compared to state-of-the-art diffusion, transformer, and deep unfolding approaches, providing an alternative to traditional deterministic methods.

**Strengths:**

The problem this work aims to solve is crucial. Recovering high-frequency details from ill-posed inverse problems is always challenging for both traditional optimization-based methods and end-to-end deterministic networks. Leveraging the powerful generative capabilities of models pre-trained on large RGB datasets and making them transferable to spectral imaging is a promising approach to address this challenge.

In addition, I like the idea of using reversible spectral unmixing to bridge spectral image and RGB image.

The novelty is enough, to address the high-frequency reconstruction challenge in spectral compressive imaging, this paper solved several problems and makes 4 key contributions: (1) the introduction of a spectral unmixing-driven predict-and-subspace refine strategy, offering improved perceptual quality and efficiency; (2) the inclusion of a reversible decomposition module, tackling the ill-posed nature of spectral unmixing; (3) focusing diffusion generation on high-frequency components, reducing training data requirements and accelerating fine-tuning; and (4) employing high-dimensional guidance with SCI imaging consistency to enhance robustness.

Moreover, this method shows nice results on the real SCI dataset, which is a hard case for existing methods.

**Weaknesses:**

In lines 296-365 of the main paper and Section A.3 of the appendix, most of the mathematical derivations are easy to follow. However, some detailed notations are unclear in the context, which reduces readability.

The motivation for designing the reversible unmixing module is not sufficiently clear. For example, why not use an SVD-based method? What advantages does the proposed unmixing module offer?

**Questions:**

(1) For the reversible unmixing module, although the experiments show that it is lightweight and efficient, I am curious how it compares with SVD, PCA, or band selection-based methods.

(2) It seems the hyperparameters, scale $s$and timestep $T$, shown in Fig. 12, are sensitive to the model's reconstruction accuracy. How do they influence the visual quality of the reconstructed image? The PSNR is not always a reliable metric for measuring image quality, especially for spectral images.

---

> ### Author Response · Authors · 2024-11-23
> **Response to Reviewer HqgD**
>
> We sincerely thank the reviewer for acknowledging the significance and novelty of our work in tackling the challenging problem of high-frequency detail reconstruction in spectral compressive imaging by leveraging pre-trained RGB generative models and introducing a reversible spectral unmixing strategy, which demonstrates promising results especially on difficult real-world SCI datasets. Please see below for our point-by-point responses to your comments.
>
> - **Comment**：In lines 296-365 of the main paper and Section A.3 of the appendix, most of the mathematical derivations are easy to follow. However, some detailed notations are unclear in the context, which reduces readability. The motivation for designing the reversible unmixing module is not sufficiently clear. For example, why not use an SVD-based method? What advantages does the proposed unmixing module offer? For the reversible unmixing module, although the experiments show that it is lightweight and efficient, I am curious how it compares with SVD, PCA, or band selection-based methods.
>
> >    **@**: We have revised the appendix and added more detailed explanations to clarify the connections between each equation, making the entire process clearer and easier to follow. Additionally, the original intent behind designing the reversible unmixing module was to achieve dimensionality reduction of spectral information channels while ensuring accurate representation of spectral data and compatibility with the RGB space. Thus, we considered using an SVD-based approach [1] for decomposition from the beginning. Our comparison results on KAIST dataset show that, compared to using the SVD method as described in the HIR-Diff paper, our learnable unmixing module achieves superior reconstruction performance and inference speed.
> >
> >| Method                | Scene 1      | Scene 2      | Scene 3      | Scene 4      | Scene 5      | Scene 6      | Scene 7      | Scene 8      | Scene 9      | Scene 10     | Average              |
> >| --------------------- | ------------ | ------------ | ------------ | ------------ | ------------ | ------------ | ------------ | ------------ | ------------ | ------------ | -------------------- |
> >| **SVD-band-select** (HIR-Diff [1]) | 40.65 0.9810 | 36.25 0.9667 | 39.20 0.9795 | 51.10 0.9948 | 38.70 0.9909 | 40.60 0.9873 | 43.12 0.9862 | 40.80 0.9867 | 33.00 0.9623 | 46.60 0.9913 | 41.00 0.9827  |
> >| **URSe**(Ours)    | 50.54 0.9982 | 54.03 0.9990 | 49.61 0.9939 | 57.02 0.9994 | 49.28 0.9984 | 49.99 0.9984 | 47.75 0.9948 | 49.05 0.9977 | 50.57 0.9967 | 50.97 0.9988 | **50.88** **0.9975** |
> >
> >
> >    Specifically, SVD-based methods generally exhibit inferior reconstruction quality compared to our designed module. The reason for this is that SVD models the relationships between spectral bands as simple linear transformations, whereas, in reality, most spectral relationships are inherently nonlinear. Although SVD provides an option of spectral decomposition, its speed advantage diminishes in multispectral image. Conversely, our learnable module not only maintains competitive inference speed but also achieves superior compression and reconstruction performance.
>
> - **Comment**: It seems the hyperparameters, scale sand timestep T, shown in Fig. 12, are sensitive to the model's reconstruction accuracy. How do they influence the visual quality of the reconstructed image? The PSNR is not always a reliable metric for measuring image quality, especially for spectral images.
>
> > **@**: We appreciate your insight regarding the sensitivity of hyperparameters **s** (scale) and **T** (timestep). It is worth noting that such parameters are inherently sensitive in most diffusion-based models, influencing generative performance and reconstruction quality. Recognizing that PSNR alone may not fully capture visual quality, particularly for spectral images, we have added **visual hyperparameter maps** in the revised appendix to supplement the PSNR and SSIM maps in the original submission. These visualizations illustrate how different hyperparameter settings affect perceptual quality, highlighting improvements or degradations that numerical metrics alone cannot capture, thus offering a more comprehensive understanding of their impact.
>
> [1] Pang, Li, et al. "HIR-Diff: Unsupervised Hyperspectral Image Restoration Via Improved Diffusion Models." Proceedings of the IEEE/CVF Conference on Computer Vision and Pattern Recognition. 2024.

---

> > ### Comment · Reviewer_HqgD · 2024-11-26
> > **Response to the author**
> >
> > The authors have addressed my confusion.

---

> > > ### Author Response · Authors · 2024-11-27
> > > **Response to Reviewer HqgD**
> > >
> > > Dear Reviewer HqgD,
> > >
> > > Thank you for your positive feedback. We're pleased that our revisions have successfully addressed your concerns. We appreciate your thoughtful review and are glad that our explanations have clarified any confusion.
> > >
> > > Kind regards,
> > >
> > > The Authors

---

### Official Review · Reviewer_wQ83 · 2024-10-29

**Soundness:** 3
**Presentation:** 3
**Contribution:** 3
**Rating:** 8
**Confidence:** 5

**Summary:**

This paper proposes a new two-stage method, namely a predict-and-subspace refine (PSR-SCI) framework, using the diffusion model for snapshot compressive imaging reconstruction. In the first stage, an inexpensive predictor generates a rough MSI approximation. Then in the second stage, a spectral unmixing-driven subspace learning module is employed to reduce the dimensionality. Eventually, a diffusion model is fine-tuned to enhance the high-frequency details. There are three technical contributions:

(i) Given the complexities of spectral unmixing models, a reversible decomposition network is introduced to implement spectral subspace learning while maintaining high reversibility.

(ii) The diffusion generation is facilitated by low- and high-frequency decomposition.

(iii) A high-dimensional guidance strategy is introduced for fine-tuning subspace diffusion, enhancing the effectiveness of guidance within the subspace.

**Strengths:**

(i) The novelty is good and interesting. The big idea of partitioning the reconstruction into two stages is cool. It seems like the latent space diffusion models. The first stage of prediction and decomposition is like the embedding process of VAE and this process is highly reversible. Besides, using the frequency domain decomposition (low- and high-frequency) to facilitate the generation process is also very fancy. It matches the nature of spectra. It is very exciting to have this work in the community of SCI reconstruction.

(ii) The performance is solid. As compared in Table 1, Figures 6, 8, and 9. The proposed method PSR-SCI not only outperforms the state-of-the-art end-to-end methods by large margins but also achieves better visual results. Compared to the SOTA diffusion-based method DiffSCI, the proposed PSR-SCI is 2.86 dB higher! It is great progress of the diffusion-based method in SCI. This work even contains the results of Pseudo-RGB on the RGB-to-HSI reconstruction task.

(iii) The presentation is well-dressed. The figure of the pipeline looks clear. I can easily see the workflow of the proposed framework. The table of quantitative comparisons is also neat. The style of this table is from the series work of MST, MST++, CST, DAUHST, etc. I like it very much. The writing also looks good and easy to follow.

(iv) Code and the reconstruction results have been submitted,

**Weaknesses:**

(i) Some modifications and explanations should be added. For example, why do you want to use an inexpensive predictor to produce an initial HSI instead of directly reconstructing the HSI results from the noisy measurements? Besides, since the process looks like the Stablediffusion - latent space diffusion. So I think it is better to discuss your work with it to highlight the differences and your contributions.

(ii) The experiments are insufficient now. The ablation study only has a visual study (Figure 10). This is far from satisfactory. I suggest you add a break-down ablation like MST, MST++, CST, and DAUHST series works to demonstrate the effectiveness of each technical contribution. Or your claim of your work - the three modules are not convincing since their effects are unknown.

(iii) The computational cost and memory usage of different methods are not reported in Table 1. I cannot judge the efficiency of the proposed method and compare it with other algorithms. I suggest you list the number of parameters and the floating-point operations per second (FLOPS).

**Questions:**

Some parts may need more explanation. For example,

In Algorithm1 (Line 270 - 287), the return value $\mathcal{A}_{diff}^h$ is fed into the VAE decoder to obtain the final reconstructed hyperspectral images. However, for the latent space diffusion, the VAE is trained to decode normal RGB images. Can it directly work on hyperspectral images? Or did you fine-tune it or something like that?

---

> ### Author Response · Authors · 2024-11-23
> **Response to Reviewer wQ83**
>
> Thank you very much for your highly positive feedback; we are delighted that you find our novel two-stage reconstruction framework both exciting and innovative, recognize its solid performance with significant advancements in diffusion-based methods for SCI reconstruction, and appreciate the clear and well-structured presentation of our work. Please see below for our point-by-point responses to your comments.
>
> - **Comment**:  Some modifications and explanations should be added. For example, why do you want to use an inexpensive predictor to produce an initial HSI instead of directly reconstructing the HSI results from the noisy measurements? Besides, I think it is better to discuss your work with latent space diffusion to highlight the differences and your contributions.
> >
> >   **@**: Thank you for the insightful suggestions. We opted to use an inexpensive predictor to produce an initial hyperspectral image (HSI) rather than directly reconstructing the HSI from noisy measurements because directly generating the spectral image from noise makes it challenging to leverage the prominent low-frequency information present in the measurements. In our two-stage framework, we recover the low-frequency and high-frequency components separately in two distinct stages. This approach allows the diffusion model to focus on high-frequency reconstruction, which reduces the number of diffusion steps needed and speeds up sampling.
> >
> >   Compared to generating HSIs entirely from noise, our two-stage approach achieves superior reconstruction quality with the same computational cost. The predictor efficiently estimates the low-frequency components, allowing the diffusion model to refine the high-frequency details effectively.
> >
> >   We have added this analysis to the ablation study section to further clarify these design decisions and highlight the advantages of our method. Additionally, we included a discussion on latent space diffusion to better differentiate our contributions and emphasize the benefits of our approach.
>
> - **Comment**: The experiments are insufficient now. The ablation study only has a visual study (Figure 10). This is far from satisfactory. I suggest you add a break-down ablation like MST, MST++, CST, and DAUHST series works to demonstrate the effectiveness of each technical contribution. Or your claim of your work - the three modules are not convincing since their effects are unknown.
> >
> >    **@**：We conducted additional ablation experiments to investigate this issue, and the results are presented in the table below. From the table, we observe the following: when the diffusion model is removed from our PSR-SCI and only the initial predictor is used, the PSNR is **37.21 dB**. When the pre-trained diffusion model, trained on a large RGB dataset, is used alone, the PSNR drops to **33.42 dB**. This reduction indicates that the pre-trained diffusion model possesses inherent generative capabilities (as evidenced by the PSNR of **33.42 dB**); however, its mismatch with the spectral imaging domain explains why the PSNR is lower compared to using the initial predictor.
> >
> >    According to our designed framework, after fine-tuning on a small spectral imaging dataset, the enhanced generative capabilities of the model become evident, resulting in an improvement of **2.83 dB** in PSNR compared to directly using the diffusion model baseline. Additionally, our URSe module reduces inference time from **312.43 s** to **13.79 s**. The use of frequency decomposition yields a further improvement of **0.47 dB**.
> >
> >    **These results suggest that although the pre-trained diffusion model trained on a large RGB dataset shows inherent generative potential, fine-tuning, spectral embedding, and targeted generation of high-frequency details are essential to optimize its performance and efficiency in spectral imaging applications.**
> >    | Initial Predictor | Freq-decomposition | URSe | Diffusion | PSNR↑     | SSIM↑     | LPIPS↓      | Best PSNR inference time (s) |
> | ----------------- | ------------------ | ---- | --------- | --------- | --------- | ----------- | ---------------------------- |
> | √                 | ×                  | ×    | ×         | 37.21     | 0.959     | 0.05718     | 0.26s                        |
> | ×                 | ×                  | ×    | √         | 33.42     | 0.883     | 0.06423     | 312.43s                      |
> | √                 | ×                  | √    | √         | 36.25     | 0.940     | 0.05375     | 13.79s                       |
> | √                 | √                  | ×    | √         | 37.67     | 0.962     | 0.04246     | 193.21s                      |
> | √                 | √                  | √    | √         | **38.14** | **0.967** | **0.02844** | 12.90s                       |

---

> > ### Author Response · Authors · 2024-11-23
> > **Response to Reviewer wQ83 (continued)**
> >
> > - **Comment**: The computational cost and memory usage of different methods are not reported in Table 1. I cannot judge the efficiency of the proposed method and compare it with other algorithms.
> >
> > >    **@**: We appreciate your concern regarding the omission of computational cost and memory usage in Table 1, which are important factors for evaluating the efficiency of different methods. To address this, we have prepared a table that details the  **inference time** for our proposed method compared to other diffusion-based algorithm:
> > >
> > >    | Method    | Category  | Reference | Inference Time (50 steps, Best for PSR-SCI) | Inference Time (600 steps, Best for DiffSCI) |
> > | --------- | --------- | --------- | ------------------------------------------- | -------------------------------------------- |
> > | DiffSCI   | Diffusion | CVPR 2024 | 130.15s                                     | 865.81s
> > | PSR-SCI-D | Diffusion | Ours      | **12.90s**                                      | **74.76s**                                     |
> > >
> > >    Our design of the **URSe module** and the two-stage pipeline enables our method to achieve superior inference speed and reduced computational load compared to other diffusion-based approaches. The initial predictor in our PSR-SCI framework is lightweight, and the proposed spectral embedding is both small and efficient. Similar to most stable diffusion or latent diffusion-based methods, the majority of our parameters are inherited from the pre-trained diffusion models we utilize.
> >
> > - **Comment**: Some parts may need more explanation. For example, In Algorithm1 (Line 270 - 287), the return value $\mathcal{A}_{\text{diff}}^h$ is fed into the VAE decoder to obtain the final reconstructed hyperspectral images. However, for the latent space diffusion, the VAE is trained to decode normal RGB images. Can it directly work on hyperspectral images? Or did you fine-tune it or something like that?
> >
> > > **@**: Yes, the VAE decoder can directly work with the return value $\mathcal{A}_{\text{diff}}^h$ to reconstruct hyperspectral images, even though it was originally trained on normal RGB images. Initially, without any fine-tuning, the compressed reconstruction achieves a **PSNR of 42.37 dB**. However, because high-frequency images contain fewer low-frequency components, the VAE can more effectively utilize its encoding space to represent high-frequency information.
> > To enhance the reconstruction quality, we fine-tuned the VAE specifically on high-frequency images. After this fine-tuning, the VAE achieved a **PSNR of 48.65 dB** on the high-frequency parts of the ground truth images. This significant improvement demonstrates that while the VAE can directly handle hyperspectral data to some extent, fine-tuning it on high-frequency components substantially boosts its performance.
> > >
> > > **In summary, although the VAE decoder trained on RGB images can be applied to hyperspectral images, fine-tuning it ensures more accurate and high-quality reconstructions. This adaptation allows the VAE to better capture the spectral nuances of hyperspectral data, leading to superior results in our framework.**

---

### Official Review · Reviewer_igVu · 2024-11-03

**Soundness:** 3
**Presentation:** 4
**Contribution:** 3
**Rating:** 6
**Confidence:** 4

**Summary:**

The authors present a novel framework named PSR-SCI aimed at enhancing CASSI reconstruction. The framework consists of a initial predictor(MST or DAUHST), a unmixing-driven reversible spectral embedding module to decomposes the MSI into subspace images and spectral coefficients, and a pre-trained RGB latent diffusion models (SD-2.1) refinement processes. Furthermore, the authors implement a high-dimensional guidance mechanism that ensures imaging consistency. Experiments on the KAIST, NTIRE, ICVL, and Harvard dataset demonstrate that PSR-SCI enhances visual quality and achieves PSNR and SSIM metrics.

---
After rebuttal: I read the authors response and have raised my score.

**Strengths:**

1. The proposed method has higher performance than previous diffusion-based methods on CASSI reconstruction.
2. The method costs less time in both training and inference time than previous diffusion-based methods.

**Weaknesses:**

1. The novelty of this approach is limited, as spectral decomposition and refinement using pre-trained RGB diffusion models have previously been introduced in the spectral image restoration field [1].
2. The method relies heavily on the initial prediction network. Although the authors claim robustness for the proposed framework, it may not perform well on unseen optical setups (such as different masks, noise, and number of channels). The framework depends on initial reconstruction results from a pre-trained CASSI reconstruction network. If the pre-trained network does not perform adequately on a new system with different optical settings (a common issue in CASSI reconstruction) the framework’s effectiveness diminishes. For example, PSR-SCI-T (initial predictor: MST) and PSR-SCI-D (initial predictor: DAUHST) demonstrate this, with weaker predictors leading to inferior reconstruction results.
3. The experiments presented are insufficient. The proposed method should be evaluated with recent state-of-the-art predictors, such as [2], DPU-9stg, and SSR-L. Additionally, comparisons with similar 'refinement' frameworks, such as [3], would further validate its performance.

[1] Pang, Li, et al. "HIR-Diff: Unsupervised Hyperspectral Image Restoration Via Improved Diffusion Models." Proceedings of the IEEE/CVF Conference on Computer Vision and Pattern Recognition. 2024.
[2] Yao, Zhiyang, et al. "SPECAT: SPatial-spEctral Cumulative-Attention Transformer for High-Resolution Hyperspectral Image Reconstruction." Proceedings of the IEEE/CVF Conference on Computer Vision and Pattern Recognition. 2024.
[3] He, Wei, et al. "An interpretable and flexible fusion prior to boost hyperspectral imaging reconstruction." Information Fusion (2024): 102528.

**Questions:**

1. Why are the SSIM metrics omitted in Table 2?
2. The number of abundance maps A in spectral unmixing should correspond to the number of endmembers. Why is an up-sampled 3-channel feature map used instead? Is this solely for compatibility with the RGB pre-trained model? The URSe module seems more like a data dimension reduction or latent space encoding operation, with little relevance to spectral unmixing

---

> ### Author Response · Authors · 2024-11-23
> **Response to Reviewer igVu**
>
> Thank you for your positive feedback; we are glad that you acknowledge our method's higher performance and reduced training and inference times compared to previous diffusion-based approaches in CASSI reconstruction. Please see below for our point-by-point responses to your comments.
>
> - **Comment**: Comparsion with the spectral decomposition and refinement using pre-trained RGB diffusion models  introduced in [1].
> >    **@**: There are four differences between our work and [1]:
> >
> >    1. **Spectral decomposition**: Our work uses a learnable spectral embedding, whereas [1] selects 3 bands directly, our spectral decomposition gets higher reconstruction quality than SVD used in [1] (50.88dB vs. 41dB, 10 kaist scenes averaged), also with much faster speed (0.0009s vs. 0.001215s).
> >    2. **Refinement design**: Our diffusion refinement design is specifically tailored for spectral images with a trainable framework, whereas [1] directly applies RGB diffusion without addressing the crucial spectral differences between RGB and multispectral images. As a result, our PSR-SCI achieves **38.14 dB**, compared to **35.23 dB** for [1].
> >    3. **Specific tasks**: The tasks (applications) are also different—our work focuses on reconstructing **N bands from 1 band**, while [1] deals with **N bands to N bands**. Our task is **more difficult** due to the **low sampling rate**.
> >    4. **Framework flexibility and performance**: Our work goes far beyond [1], offering a more flexible framework and significantly better performance (38.14dB vs. 35.23dB). We implemented the idea of [1] for SCI comparison, even though [1] was not designed for SCI.
> >
> >    Specifically, firstly, [1] is not trainable; it only uses a pre-trained RGB diffusion model and cannot be fine-tuned on spectral images. This makes the diffusion model's generated results still deviate from the spectral images due to the wavelengths of RGB images and spectral images being quite different. RGB imaging typically captures red (650 nm), green (560 nm), and blue (460 nm) wavelengths, whereas spectral imaging spans a broader range of wavelengths, extending into the near-infrared (750–2500 nm). We solved this problem by designing a fine-tunable diffusion framework, which is fine-tuned on a spectral image dataset, making full use of a pre-trained RGB diffusion model but also making it suitable for the spectral imaging task.
> >
> >    Secondly, the introduced diffusion method [1] is specifically designed for denoising or restoration of spectral images, focusing on the restoration from N bands to N bands. In contrast, our task involves reconstructing multispectral images from a single-shot snapshot compressed image (1 band to N bands), which is significantly more challenging due to the low sampling rate. These two tasks face fundamentally different challenges: denoising involves removing noise, while compressed sensing aims to recover full-spectral-spatial spectral image given a under-sampled measurement.
> >
> >    Meanwhile, our spectral compression and decomposition process reduces spectral images to three channels using trainable modules, achieving high reconstruction quality (41dB vs. 50.88dB, 10 kaist scenes averaged), also with much faster speed (0.0009s vs. *0.001215s).
> >
> >    Although [1] was not originally designed for the Snapshot Compressive Imaging (SCI) task we address in this paper, for a more comprehensive comparison, we incorporated it into the second stage of our two-stage framework (while keeping the first stage unchanged). We compared HIR-Diff with our proposed PSR-SCI, and the reconstruction performance for snapshot tasks is summarized below. The results indicate that HIR-Diff achieves reasonable reconstruction quality, but its performance metrics lag behind state-of-the-art methods. Our approach, leveraging guidance and additional trainable modules, demonstrates clear advantages over existing diffusion methods for this task, such as DiffSCI, in terms of performance metrics, visual reconstruction quality, and inference speed.
> >.
> >| Method                                                     | Reference | PSNR  | SSIM  |
> >| ---------------------------------------------------------- | --------- | ----- | ----- |
> >| HIR-DIff (+our fisrt stage-to ensure it works for SCI) [1] | CVPR-2024 | 35.23 | 0.959 |
> >| PSR-SCI-D                                                  | Ours      | **38.14** | **0.967** |

---

> > ### Author Response · Authors · 2024-11-23
> > **Response to Reviewer igVu (continued)**
> >
> > - **Comment**:  The authors claim robustness for the proposed framework, it may not perform well on unseen optical setups (such as different masks, noise, and number of channels).
> > >    **@**: While our method does utilize an initial prediction network, we have demonstrated its robustness across different initial predictors and optical setups. As evidenced in our summaried ablation studies below (also see Table 1 and Table 2 in our manuscript), using MST as the initial predictor, our PSR-SCI framework achieves improvements of **1.87 dB** on the KAIST dataset and **3 dB** on the ICVL dataset. With DAUHST as the initial predictor, we observe gains of **0.93 dB** and **2.39 dB**, respectively. Given that the PSNR values are approximately **40 dB**, these improvements are substantial. The marginally smaller gains with higher PSNRs are expected due to the logarithmic nature of the metric, which makes further improvements progressively more challenging as quality increases.
> > >
> > >In summary, our framework consistently reconstructs high-frequency details regardless of the initial predictor used. When encountering imaging systems with new optical setups—such as different masks, noise levels, or channel numbers—we simply replace the initial predictor with a reconstruction model tailored to that system. This ensures our framework remains effective even if the pre-trained network wasn't designed for those settings. Training a new predictor is only necessary for entirely new imaging systems, which is standard practice. While most deep learning-based spectral reconstruction methods require retraining for different setups, our framework offers stable performance across various predictors and datasets, demonstrating robustness and adaptability beyond reliance on the initial prediction network.
> > >
> > - **Comment**: The experiments presented are insufficient. The proposed method should be evaluated with recent state-of-the-art predictors, such as [2], DPU-9stg, and SSR-L.
> > >
> >  > **@**: We acknowledge your concern about the sufficiency of our experiments and the need to evaluate our method with recent state-of-the-art predictors such as [2], DPU-9stg, and SSR-L.
> >  >
> >  > To address your suggestion, we have added comparisons with these predictors on the **ICVL, NTIRE, Harvard** datasets. The results are as follows:
> >  >
> >  > | Dataset | Metric  | DAUHST-3stg (NeurIPS 2022) | MST-L (CVPR 2022) | DPU-9stg (CVPR 2024) | SSR-L (CVPR 2024) | DiffSCI (CVPR 2024) | PSR-SCI-D | PSR-SCI-DPU | PSR-SCI-SSR |
> >  > | ------- | ------- | -------------------------- | ----------------- | -------------------- | ----------------- | ------------------- | --------- | ----------- | ----------- |
> >  > | ICVL    | PSNR↑   | 34.64                      | 34.03             | 36.56                | 36.25             | 33.02               | 37.03     | 37.25       | 37.14       |
> >  > |         | SSIM↑   | 0.890                      | 0.885             | 0.918                | 0.914             | 0.868               | 0.918     | 0.923       | 0.918       |
> >  > |         | MANIQA↑ | 0.200                      | 0.209             | 0.200                | 0.209             | 0.207               | 0.217     | 0.216       | 0.213       |
> >  > | NTIRE   | PSNR↑   | 34.44                      | 33.04             | 36.25                | 35.44             | 32.79               | 36.44     | 36.62       | 36.53       |
> >  > |         | SSIM↑   | 0.927                      | 0.914             | 0.945                | 0.942             | 0.903               | 0.953     | 0.955       | 0.948       |
> >  > |         | MANIQA↑ | 0.214                      | 0.210             | 0.226                | 0.230             | 0.205               | 0.233     | 0.238       | 0.240       |
> >  > | Harvard | PSNR↑   | 25.57                      | 24.01             | 27.05                | 25.93             | 24.68               | 26.90     | 28.58       | 29.02       |
> >  > |         | SSIM↑   | 0.622                      | 0.594             | 0.650                | 0.597             | 0.602               | 0.776     | 0.764       | 0.728       |
> >  > |         | MANIQA↑ | 0.187                      | 0.204             | 0.197                | 0.195             | 0.174               | 0.205     | 0.239       | 0.247       |
> >  >
> >  >
> >  > The observations from the added table clearly indicate that, with DAUHST, DPU-9stg, and SSR-L as our initial predictors, our PSR-SCI method shows noticeable improvements in terms of PSNR, SSIM, and MANIQA. For example, on the ICVL dataset, PSR-SCI achieves a **2.39 dB** improvement over DAUHST-3stg, **0.69 dB** over DPU-9stg, and **0.89 dB** over SSR-L. Importantly, the improvements are more significant when the initial predictor's PSNR is lower, which is expected because higher initial PSNR values leave less room for further enhancement. This consistent performance boost underscores the effectiveness of our method across various predictors.

---

> > > ### Author Response · Authors · 2024-11-23
> > > **Response to Reviewer igVu (continued)**
> > >
> > > * **Comment**: Additionally, comparisons with similar 'refinement' frameworks, such as [3], would further validate its performance.
> > > >
> > > > **@**：We appreciate your suggestion to compare our method with similar refinement frameworks, such as DAUHST-SP2 [3], to further validate its performance. In response, we have added a comparison with the latest refinement-based model DAUHST-SP2 to Table 1 in our manuscript. For a fair evaluation, we used DAUHST as the first-stage predictor for both DAUHST-SP2 and our proposed PSR-SCI method. The results are as follows (at the time of the rebuttal, the code for [3] had not yet been released; we cite the PSNR and SSIM reported in the paper):
> > > >
> > > > | Refinement-based Algorithms                 | Category        | Reference               | PSNR↑     | SSIM↑     | LPIPS↓      | MUSIQ↑    | MANIQA↑    | CLIP-IQA↑  |
> > > > | ------------------------------------------- | --------------- | ----------------------- | --------- | --------- | ----------- | --------- | ---------- | ---------- |
> > > > | DAUHST-SP2 (refined on DAUHST) [3]          | Subspace  prior | Information Fusion 2024 | 37.61     | 0.966     | -           | -         | -          | -          |
> > > > | DiffSCI                                     | Diffusion       | CVPR 2024               | 35.28     | 0.916     | 0.07421     | 39.64     | 0.23598    | 0.3315     |
> > > > | **PSR-SCI-D (DAUHST as initial predictor)** | Diffusion       | Ours                    | **38.14** | **0.967** | **0.02844** | **42.73** | **0.2527** | **0.3561** |
> > > >
> > > > Our observations indicate that **PSR-SCI achieves a PSNR of 38.14 dB**, outperforming **DAUHST-SP2's 37.61 dB**. Furthermore, PSR-SCI demonstrates superior performance over DAUHST-SP2 in terms of **SSIM**, **LPIPS**, **MUSIQ**, and **MANIQA** metrics. These improvements highlight the effectiveness of our method in enhancing reconstruction quality compared to existing refinement frameworks.
> > > >
> > > > By including this comparison, we aim to substantiate the advantages of our approach over similar models. The consistent improvements across multiple evaluation metrics reinforce the robustness and efficacy of PSR-SCI in spectral image reconstruction.

---

> > > > ### Author Response · Authors · 2024-11-23
> > > > **Response to Reviewer igVu (continued)**
> > > >
> > > > - **Comment**: why are the SSIM metrics omitted in Table 2?
> > > > >
> > > > >    **@**: We appreciate your observation regarding the omission of the Structural Similarity Index Measure (SSIM) in Table 2 of our manuscript. SSIM is indeed a vital metric for assessing the similarity between the original and reconstructed images, and we included SSIM comparisons across different methods in following table to highlight this aspect.
> > > > >
> > > > >    In Table 2, we aimed to focus on the  **MANIQA**, a metric specifically designed to evaluate the perceptual quality of images reconstructed by diffusion-based methods. MANIQA is particularly important for our work because it captures human visual perception more effectively than traditional metrics like SSIM, especially in the context of diffusion models that prioritize perceptual fidelity.
> > > > >
> > > > >    Due to page limitations imposed, we replaced SSIM with MANIQA in Table 2 to present a broader range of evaluation metrics without exceeding the allowed length. However, recognizing the importance of SSIM in providing a comprehensive evaluation, we have included the previously omitted SSIM values from Table 2 in the table below:
> > > > >
> > > > >    | Dataset | Metric    | DAUHST-3stg (NeurIPS 2022) | MST-L (CVPR 2022) | DPU-9stg (CVPR 2024) | SSR-L (CVPR 2024) | DiffSCI (CVPR 2024) | PSR-SCI-D (Ours) |
> > > > >    | ------- | --------- | -------------------------- | ----------------- | -------------------- | ----------------- | ------------------- | ---------------- |
> > > > >    | ICVL    | PSNR↑     | 34.64                      | 34.03             | 36.56                | 36.25             | 33.02               | **37.03**        |
> > > > >    |         | **SSIM↑** | 0.890                      | 0.885             | **0.918**            | 0.914             | 0.868               | **0.918**        |
> > > > >    |         | MANIQA↑   | 0.200                      | 0.209             | 0.200                | 0.209             | 0.207               | **0.217**        |
> > > > >    | NTIRE   | PSNR↑     | 34.44                      | 33.04             | 36.25                | 35.44             | 32.79               | **36.44**        |
> > > > >    |         | **SSIM↑** | 0.927                      | 0.914             | 0.945                | 0.942             | 0.903               | **0.953**        |
> > > > >    |         | MANIQA↑   | 0.187                      | 0.204             | 0.197                | 0.195             | 0.174               | **0.205**        |
> > > > >    | Harvard | PSNR↑     | 25.57                      | 24.01             | **27.05**            | 25.93             | 24.68               | 26.90            |
> > > > >    |         | **SSIM↑** | 0.622                      | 0.594             | 0.650                | 0.597             | 0.602               | **0.776**        |
> > > > >    |         | MANIQA↑   | 0.214                      | 0.210             | 0.226                | 0.230             | 0.205               | **0.233**        |
> > > > >
> > > > >    From the updated results, it is evident that our proposed PSR-SCI method achieves the best SSIM scores, aligning with the superior performance indicated by both **PSNR** and **MANIQA** metrics. This consistency across multiple evaluation criteria reinforces the effectiveness of our method in delivering high-quality reconstructions that are both structurally accurate and perceptually pleasing.

---

> > > > > ### Author Response · Authors · 2024-11-23
> > > > > **Response to Reviewer igVu (continued)**
> > > > >
> > > > > - **Comment**: The number of abundance maps A in spectral unmixing should correspond to the number of endmembers. Why is an up-sampled 3-channel feature map used instead? Is this solely for compatibility with the RGB pre-trained model?
> > > > > >
> > > > > >   **@**：Our **Unmixing-driven Reversible Spectral Embedding module (URSe)** is indeed inspired by spectral unmixing, aligning with its decomposition principles. However, we intentionally set the number of channels to **three** rather than matching the number of endmembers. This choice is not solely for compatibility with RGB pre-trained models but serves a strategic purpose.
> > > > > >
> > > > > >   By constraining the spectral data to three channels, we can effectively leverage powerful image priors from models like Stable Diffusion (SD), which are trained on large RGB image datasets. This integration enhances reconstruction quality by utilizing rich visual features learned from extensive data.
> > > > > >
> > > > > >   The upsampling step addresses the difference in spatial dimensions between our multispectral images (MSI) and the SD model's requirements. Our MSI data has a resolution of **256**, while the SD model operates on images of size **512 × 512**. Upsampling ensures that our data aligns with the SD model's input size, allowing seamless integration and optimal performance.
> > > > > >
> > > > > >   While the URSe module performs dimensionality reduction similar to latent space encoding, it remains closely related to spectral unmixing in its conceptual foundation. The module is designed to be reversible, preserving essential spectral information for accurate reconstruction.
> > > > > >
> > > > > >   In essence, while our URSe module does not perform spectral unmixing in the traditional sense (due to the fixed three-channel output), it is unmixing-driven in its design philosophy. The module bridges the gap between high-dimensional spectral data and the three-channel input expected by powerful pre-trained models. This design choice enhances the reconstruction performance by combining spectral decomposition concepts with advanced image priors.

---

> ### Author Response · Authors · 2024-11-26
> **Response to Reviewer igVu**
>
> Dear Reviewer igVu,
>
> We, the authors, sincerely appreciate you taking the time to read our response and for raising your score. Your support is incredibly encouraging and serves as significant inspiration for all of us working on this paper.
>
> Thank you once again for your valuable feedback.
>
>
>
> Kind regards,
>
> The Authors

---

### Meta-Review · Area_Chair_H9SP · 2024-12-14

**Metareview:**

The paper proposed a Predict-and-unmixing-driven-Subspace-Refine framework (PSR-SCI), which achieves higher performance than previous diffusion-based methods. The motivation is clear and interesting.
Given that all four reviewers have provided high evaluations and the authors have responded in great detail to the reviewers’ comments, which has further improved the quality of the paper, I recommend accepting this paper.

**Additional Comments On Reviewer Discussion:**

The authors provided detailed responses and there is no additional comments.

---

### Decision · Program_Chairs · 2025-01-22

Accept (Spotlight)